# Pharmacological inhibition of lysine-specific demethylase 1 (LSD1) induces global transcriptional deregulation and ultrastructural alterations that impair viability in *Schistosoma mansoni*

Vitor Coutinho Carneiro[1¤], Isabel Caetano de Abreu da Silva[1], Murilo Sena Amaral[2], Adriana S. A. Pereira[2,3], Gilbert Oliveira Silveira[2,3], David da Silva Pires[2], Sergio Verjovski-Almeida[2,3], Frank J. Dekker[4], Dante Rotili[5], Antonello Mai[5], Eduardo José Lopes-Torres[6], Dina Robaa[7], Wolfgang Sippl[7], Raymond J. Pierce[8], M. Teresa Borrello[9], A. Ganesan[10], Julien Lancelot[8], Silvana Thiengo[11], Monica Ammon Fernandez[11], Amanda Roberta Revoredo Vicentino[1], Marina Moraes Mourão[12], Fernanda Sales Coelho[12], Marcelo Rosado Fantappié[1]*

1 Instituto de Bioquímica Médica Leopoldo de Meis, Programa de Biologia Molecular e Biotecnologia, Centro de Ciências da Saúde, Universidade Federal do Rio de Janeiro, Rio de Janeiro, Brazil, 2 Laboratório de Parasitologia, Instituto Butantan, São Paulo, Brazil, 3 Departamento de Bioquímica, Instituto de Química, Universidade de São Paulo, São Paulo, Brasil, 4 Department of Chemical and Pharmaceutical Biology, University of Groningen, Antonius Deusinglaan, AV Groningen, Netherlands, 5 Department of Drug Chemistry and Technologies, Sapienza University of Rome, Rome, Italy, 6 Laboratório de Helmintologia Romero Lascasas Porto, Faculdade de Ciências Médicas, Universidade do Estado do Rio de Janeiro, Rio de Janeiro, Brazil, 7 Institute of Pharmacy, Martin Luther University of Halle-Wittenberg, Germany, 8 Université de Lille, CNRS, Inserm, CHU Lille, Institut Pasteur de Lille, U1019—UMR 9017—CIIL—Centre d'Infection et d'Immunité de Lille, Lille, France, 9 Centre de Recherche en Cancérologie de Marseille (CRCM), INSERM U1068, CNRS UMR 7258, Aix-Marseille Université and Institut Paoli-Calmettes, Parc Scientifique et Technologique de Luminy, Marseille, France, 10 School of Pharmacy, University of East Anglia, Norwich NR4 7TJ, United Kingdom, 11 Laboratório de Malacologia, Fundação Oswaldo Cruz, Instituto Oswaldo Cruz, Rio de Janeiro, Brazil, 12 Grupo de Helmintologia e Malacologia Médica, Instituto René Rachou, Fundação Oswaldo Cruz, Belo Horizonte, Brazil

¤ Current address: Division of Epigenetics, German Cancer Research Center, Heidelberg, Germany.
* fantappie@bioqmed.ufrj.br

## Abstract

Treatment and control of schistosomiasis still rely on only one effective drug, praziquantel (PZQ) and, due to mass treatment, the increasing risk of selecting for schistosome strains that are resistant to PZQ has alerted investigators to the urgent need to develop novel therapeutic strategies. The histone-modifying enzymes (HMEs) represent promising targets for the development of epigenetic drugs against *Schistosoma mansoni*. In the present study, we targeted the *S. mansoni* lysine-specific demethylase 1 (SmLSD1), a transcriptional corepressor, using a novel and selective synthetic inhibitor, MC3935, which was used to treat schistosomula and adult worms *in vitro*. By using cell viability assays and optical and electron microscopy, we showed that treatment with MC3935 affected parasite motility, egg-laying, tegument, and cellular organelle structures, culminating in the death of schistosomula and adult worms. *In silico* molecular modeling and docking analysis suggested that MC3935

**Data Availability Statement:** The raw RNA-seq data was deposited at NCBI under BioProject accession number PRJNA361136.

**Funding:** This work was supported in part by a grant from the European Union's Seventh Framework Programme under agreement no. 602080. MRF, was supported by the Coordenação de Aperfeiçoamento de Pessoal de Nível Superior (CAPES) and Conselho Nacional de Desenvolvimento Científico e Tecnológico (CNPq), Brazil. MRF, MMM, SVA are recipient of established investigator fellowship award from CNPq. Fellowships from FAPESP has supported GOS (2018/24015-0). RP received institutional support from Inserm, CNRS, Pasteur Institute of Lille and Lille University. AM was supported by PRIN 2016 (prot. 20152TE5PK) and AIRC 2016 (n. 19162) funds. W.S. and D.R. were supported by the European Regional Development Fund of the European Commission. EJLT was supported by FAPERJ JCNE (Productive Fellowship Program, grant 202.660/2018). FD was supported by the Netherlands Organization for Scientific Research (NWO), VIDI grant (723.012.005). The funders had no role in study design, data collection and analysis, decision to publish, or preparation of the manuscript.

**Competing interests:** The authors have declared that no competing interests exist.

binds to the catalytic pocket of SmLSD1. Western blot analysis revealed that MC3935 inhibited SmLSD1 demethylation activity of H3K4me1/2. Knockdown of SmLSD1 by RNAi recapitulated MC3935 phenotypes in adult worms. RNA-Seq analysis of MC3935-treated parasites revealed significant differences in gene expression related to critical biological processes. Collectively, our findings show that SmLSD1 is a promising drug target for the treatment of schistosomiasis and strongly support the further development and *in vivo* testing of selective schistosome LSD1 inhibitors.

## Author summary

Schistosomiasis mansoni is a chronic and debilitating tropical disease caused by the helminth *Schistosoma mansoni*. The control and treatment of the disease rely almost exclusively on praziquantel (PZQ). Thus, there is an urgent need to search for promising protein targets to develop new drugs. Drugs that inhibit enzymes that modify the chromatin structure have been developed for a number of diseases. We and others have shown that *S. mansoni* epigenetic enzymes are also potential therapeutic targets. Here we evaluated the potential of the *S. mansoni* histone demethylase LSD1 (SmLSD1) as a drug target. We reported the synthesis of a novel and potent LSD1 inhibitor, MC3935, and show that it selectively inhibited the enzymatic activity of SmLSD1. Treatment of juvenile or adult worms with MC3935 caused severe damage to the tegument of the parasites and compromised egg production. Importantly, MC3935 proved to be highly toxic to *S. mansoni*, culminating in the death of juvenile or adult worms within 96 h. Transcriptomic analysis of MC3935-treated parasites revealed changes in the gene expression of hundreds of genes involved in key biological processes. Importantly, SmLSD1 contains unique sequences within its polypeptide chain that could be explored to develop a *S. mansoni* selective drug.

## Introduction

Schistosomes are large metazoan pathogens that parasitize over 200 million people worldwide, resulting in up to 300,000 deaths per year [1,2]. No efficacious vaccine is available for human schistosomiasis, and the control and treatment of the disease rely almost exclusively on praziquantel (PZQ), the only effective drug against all schistosome species infecting humans. Despite its efficacy, PZQ does not kill juvenile parasites, allowing reinfection [3], and there is a constant concern with the appearance of PZQ-resistant strains of *Schistosoma* [4–6]. Thus, there is an urgent need to search for promising protein targets to develop new drugs.

Transcription factors and chromatin modifiers play primary roles in the programming and reprogramming of cellular states during development and differentiation, as well as in maintaining the correct cellular transcriptional profile [7]. Indeed, a plethora of groundbreaking studies has demonstrated the importance of posttranslational modifications of histones for transcription control and normal cell development. Therefore, the deregulation of epigenetic control is a common feature of a number of diseases, including cancer [7].

The complexity of schistosome development and differentiation implies tight control of gene expression at all stages of the life cycle and that epigenetic mechanisms are likely to play key roles in these processes. In recent years, targeting the *Schistosoma mansoni* epigenome has emerged as a new and promising strategy to control schistosomiasis. The study of histone acetylation in *S. mansoni* biology and the effect of inhibitors of histone deacetylases (HDACs and

SIRTs) or histone acetyltransferases (HATs) on parasite development and survival have demonstrated the importance of these enzymes as potential therapeutic targets [8–12].

Unlike histone lysine acetylation, which is generally coupled to gene activation, histone lysine methylation can have different biological associations depending on the position of the lysine residue and the degree of methylation [13]. Patterns of specific lysine methyl modifications are achieved by a precise lysine methylation system, consisting of proteins that add, remove and recognize the specific lysine methyl marks. Importantly, histone lysine methylation [14–16] and demethylation [17] have been recently demonstrated to be potential drug targets against *S. mansoni*.

Lysine-specific demethylase 1 (LSD1) was the first protein reported to exhibit histone demethylase activity and has since been shown to have multiple essential roles in metazoan biology [18]. LSD1 enzymes are characterized by the presence of an amine oxidase (AO)-like domain, which is dependent on its cofactor flavin-adenine dinucleotide (FAD), a SWIRM domain, which is unique to chromatin-associated proteins [19] and an additional coiled-coil TOWER domain [20]. LSD1 is a component of the CoREST transcriptional corepressor complex that also contains CoREST, CtBP, HDAC1 and HDAC2. As part of this complex, LSD1 demethylates mono-methyl and di-methyl histone H3 at Lys4 (H3K4me1/2), but not H3K4me3 [21]. In addition, when recruited by androgen or estrogen receptor, LSD1 functions as a H3K9 demethylase. Given the high level of expression and enzymatic activity of LSD1 in many types of tumors, there has been significant recent interest in the development of pharmacological inhibitors [22].

In our continuing effort to study the biology and therapeutic potential of epigenetic regulators in *S. mansoni*, we have found schistosome LSD1 (SmLSD1, Sm_150560) as a potential drug target (this paper). During the course of our investigation, a recent publication [17] described the repurposing of some anthracyclines as anti-Schistosoma agents, suggesting by *in silico* docking SmLSD1 as a putative target without any evidence of enzyme inhibition.

In the present work we show the SmLSD1-inhibitory activity of a novel synthetic LSD1 inhibitor MC3935 (100-fold more potent than the canonical human LSD1 inhibitor tranylcypromine, TCP). In addition, we show that the LSD1 inhibitor MC3935 was able to kill both adult worms and schistosomula *in vitro*. Importantly, silencing of SmLSD1 by dsRNAi partially recapitulated MC3935-treatment phenotypes in adult worms. Our RNA-Seq analysis revealed a large-scale transcriptional deregulation in parasites that were treated with sublethal doses of MC3935, which could be the primary cause of the ultrastructure defects and death of *S. mansoni*. Together, these findings elucidate the biological relevance of histone lysine methylation in *S. mansoni* and provide insights into the therapeutic potential of SmLSD1 to control schistosomiasis.

## Materials and methods

### Ethics statement

Animals were handled in strict accordance with good animal practice as defined by the Animal Use Ethics Committee of UFRJ (Universidade Federal do Rio de Janeiro). This protocol was registered at the National Council for Animal Experimentation (CONCEA- 01200.001568/2013-87) with approval number 086/14. The study adhered to the institution's guidelines for animal husbandry.

### Protein alignment and phylogenetic relationships

Multiple-sequence alignment of the full-length proteins was performed using representatives of *C. elegans* (NP_493366.1), *D. melanogaster* (NP_649194.1), *H. sapiens* (NP_055828.2), *M.*

*musculus* (NP_598633.2), *D. rerio* (XP_005158840.1), *A. thaliana* (NP_187981.1), *S. japonicum* (TNN15244.1), *S. haematobium* (XP_012793780.1), and *S. mansoni* (Smp_150560) LSD1 protein sequences, as previously published [23]. Pairwise comparisons to the reference followed by calculation of the maximum distance matrix resulted in an unrooted phylogenetic tree, which was visualized using Tree of Life v1.0 [24].

## Chemistry

Compound MC3935 (S1A Fig) was synthesized by coupling of the racemic *tert*-butyl (2-(4-aminophenyl)cyclopropyl)carbamate, prepared as previously reported [25], with the commercially available 4-ethynylbenzoic acid, followed by acidic deprotection of the Boc-protected amine.

[1]H-NMR spectra were recorded at 400 MHz using a Bruker AC 400 spectrometer; chemical shifts are reported in ppm units relative to the internal reference tetramethylsilane ($Me_4Si$). Mass spectra were recorded on an API-TOF Mariner by Perspective Biosystem (Stratford, Texas, USA). Samples were injected by a Harvard pump at a flow rate of 5–10 μL/min and infused into the electrospray system. All compounds were routinely checked by TLC and [1]H-NMR. TLC was performed on aluminum-backed silica gel plates (Merck DC, Alufolien Kieselgel 60 $F_{254}$) with spots visualized by UV light or using a $KMnO_4$ alkaline solution. All solvents were reagent grade and, when necessary, were purified and dried by standard methods. The concentration of solutions after reactions and extractions involved the use of a rotary evaporator operating at a reduced pressure of ~ 20 Torr. Organic solutions were dried over anhydrous sodium sulfate. Elemental analysis was used to determine the purity of the final compound 1 (MC3935) which was >95%. Analytical results were within 0.40% of the theoretical values. As a rule, the sample prepared for physical and biological studies was dried in high vacuum over $P_2O_5$ for 20 h at temperatures ranging from 25 to 40°C. Abbreviations are defined as follows: dimethylformamide (DMF), *N*-(3-dimethylaminopropyl)-*N'*-ethylcarbodiimide (EDCI), 1-hydroxybenzotriazole hydrate (HOBt), triethylamine (TEA), ethyl acetate (EtOAc) and tetrahydrofuran (THF).

## Preparation of tert-Butyl- (trans-2(4-(4-ethynylbenzamido)phenyl) cyclopropyl) carbamate (3)

4-Ethynylbenzoic acid (135.4 mg, 0.93 mmol, 1.15 eq), EDCI (216.2 mg, 1.13 mmol, 1.4 eq), HOBt (152.4 mg, 1.13 mmol, 1.4 eq) and TEA (0.43 mL, 3.06 mmol, 3.8 eq) were added sequentially to a solution of 2 (200 mg, 0.805 mmol, 1.0 eq) in dry DMF (4.5 mL) (S1 Fig). The resulting mixture was then stirred for approximately 7 h at room temperature and, after completion of the reaction, quenched with $NaHCO_3$ saturated solution (40 mL). The aqueous solution was extracted with EtOAc (4 x 25 mL); washed with 0.1 N $KHSO_4$ solution (2 x 10 mL), $NaHCO_3$ saturated solution (3 x 10 mL) and brine (3 x 5 mL); dried over anhydrous $Na_2SO_4$ and finally concentrated under vacuum. The crude product was then purified by column chromatography on silica gel eluting with a mixture EtOAc:hexane 25:75 to afford 3 as a pink solid (193 mg, 64%). [1]H-NMR (400 MHz, $CDCl_3$): δ 1.06–1.10 (m, 2H, -C*H₂*-), 1.39 (s, 9H, -COO (C*H₃*)₃), 1.94–1.98 (m, 1H, Ar-C*H*-), 2.63 (m, 1H, -C*H*-NH-COO(CH₃)₃), 3.16 (s, 1H, *H*C≡C-), 4.78 (s, 1H, -N*H*-COO(CH₃)₃), 7.07–7.09 (d, 2H, *H*-Ar), 7.44–7.47 (d, 2H, *H*-Ar), 7.52–7.54 (d, 2H, *H*-Ar), 7.69 (s, 1H, Ar-CO-N*H*-Ar), 7.74–7.76 (d, 2H, *H*-Ar). MS (ESI), m/z: 377 [M + H]⁺.

## Preparation of N-(4-(trans-2-aminocyclopropyl)phenyl)-4-ethynylbenzamide hydrochloride (1, MC3935)

To a solution of 3 (125 mg, 0.332 mmol, 1 eq.) in dry THF (9 mL) was added 4N HCl in dioxane (5.4 mL, 21.6 mmol, 65 eq.) while cooling at 0˚C. Then, the resulting suspension was stirred at room temperature for approximately 1 h. Finally, the suspension was filtered off and washed over the filter in sequence with dry THF (1 x 3 mL) and dry diethyl ether (4 x 3 mL) to afford 1 (MC3935) as a slightly pink solid (90.8 mg, 87.5%). $^1$H-NMR (400 MHz, DMSO-$d_6$): δ 1.16–1.21 (m, 1H, -CH$H$-), 1.33–1.38 (m, 1H, -C$H$H-), 2.26–2.31 (m, 1H, -C$H$-Ar), 2.77–2.80 (m, 1H, -C$H$-NH$_2$·HCl), 4.43 (s, 1H, $H$C≡C-), 7.14–7.16 (d, 2H, $H$-Ar), 7.63–7.65 (d, 2H, $H$-Ar), 7.70–7.72 (d, 2H, $H$-Ar), 7.95–7.98 (d, 2H, $H$-Ar), 8.28 (br s, 3H, -CH-N$H_2$·$H$Cl), 10.33 (s, 1H, Ar-CON$H$-Ar). MS (ESI), m/z: 277 [M + H]$^+$. Anal. (C$_{18}$H$_{17}$ClN$_2$O) Calcd. (%): C, 69.12; H, 5.48; Cl, 11.33; N, 8.96. Found (%) C, 69.26; H, 5.50; Cl, 11.27; N, 8.87.

### Biochemistry

Human LSD1 (KDM1A) (lysine (K)-specific demethylase 1A) was purchased from BPS Bioscience (catalog No 50097). Monomethylated histone peptide H3K4N(CH$_3$) was purchased from Pepscan and horseradish peroxidase (HRP) from Pierce (Catalog No. 31490). The reagents for buffer preparation were purchased from Merck, Netherlands. Reactions were conducted in black 96-well flat-bottom microplates (Corning). The fluorescence measurements were carried out in a Synergy H1 Hybrid Multi-Mode Microplate Reader (BioTek, USA) and the gain setting of the instrument was adjusted to 70. GraphPad Prism 5.0 was used for determination of the half-maximal inhibitory concentration (IC$_{50}$). Nonlinear regression was used for data fitting.

### Human LSD1 inhibition assay

Compound 1 (MC3935) was screened for inhibition against human recombinant LSD1. The *in vitro* assay was based on the oxidative demethylation of the monomethylated histone peptide H3K4N(CH$_3$) via a FAD/FADH$_2$ mediated reduction of O$_2$ to H$_2$O$_2$. The remaining LSD1 activity was monitored via the detection of the amount of H$_2$O$_2$ formed. This was done by horseradish peroxidase (HRP), which reduces H$_2$O$_2$ to H$_2$O using Amplex Red[2] as the electron donor. The resulting product, resorufin, was highly fluorescent at 590 nm. The inhibition assay was performed as described previously [26]. The compound was preincubated at different concentrations with LSD1 for 15 min at room temperature in the presence of HRP-Amplex Red. The substrate was then added, and the fluorescence was measured for 30 min.

### Homology modeling

The amino acid sequence of SmLSD1 was retrieved from Uniprot [27] (accession number: G4VK09). Subsequently, alignment of SmLSD1 and HsLSD1 sequences was performed using MOE version 2018.01 (*Molecular Operating Environment (MOE)*, 2018.01; Chemical Computing Group Inc., Canada). Long inserts in the sequence of SmLSD1 (aa: 205–271, 324–385, 828–860, 883–910, and 965–983) were deleted and consequently not modeled. The saved alignment file was used to generate a homology model of SmLSD1 based on the cocrystal structure of hLSD1 with MC2580 (PDB ID 2XAS) [26] using MODELLER 9.11 [28].

### Ligand preparation

Similar to the observed adduct of the analogous tranylcypromine derivative MC2584 with FAD (PDB ID 2XAQ [29]), an N5 adduct of MC3935 with FAD was generated in MOE using

only the flavin ring of FAD. The generated adduct was then cured using LigPrep (Schrödinger Release 2018–1): protonation states were assigned at pH 7±1 using Epik, tautomeric forms, as well as possible conformers were generated, and energy minimized using the OPLS03 force field. As a result, 25 low-energy conformers were generated using the bioactive search module implemented in Schrödinger.

### Protein preparation

The generated homology model of SmLSD1 was prepared with Schrödinger's Protein Preparation Wizard (Schrödinger Release 2018–1); where hydrogen atoms were added and the hydrogen bond network was optimized. The protonation states at pH 7.0 were predicted using the PROPKA tool in Schrödinger, and the structure was subsequently subjected to a restrained energy minimization using the OPLS03 force field (RMSD of the atom displacement for terminating the minimization was 0.3 Å).

### Docking

The receptor grid preparation for the docking procedure was carried out by assigning the coordinates of the cut cocrystallized adduct (only the flavin ring was kept in FAD) as the centroid of the grid box. Molecular docking was performed using Glide (Schrödinger Release 2018–1) in the Standard Precision mode. A total of 20 poses per ligand conformer were included in the postdocking minimization step, and a maximum of one docking pose was stored for each conformer.

### Parasite stock

The Belo Horizonte strain of *Schistosoma mansoni* (Belo Horizonte, Brazil) was maintained in the snail (*Biomphalaria glabrata*) as the intermediate host and the golden hamster (*Mesocricetus auratus*) as the definitive host [30]. Female hamsters aged 3–4 weeks, weighing 50–60 g, were infected by exposure to a *S. mansoni* cercarial suspension containing approximately 250 cercariae using intradermal injection. The adult worms were obtained by hepatoportal perfusion at 42–49 days postinfection. Cercariae were released from infected snails and mechanically transformed to obtain schistosomula *in vitro* [31].

### Treatment of *S. mansoni* with LSD1 inhibitors

Schistosomula or adult worms were treated with different concentrations of LSD1 inhibitors, as indicated in the figure legends. For each treatment condition, 10 worm pairs were maintained in 60-mm diameter culture dishes in 2 mL of culture medium (medium M169 (Gibco) supplemented with 10% fetal bovine serum (Vitrocell), penicillin/streptomycin, amphotericin and gentamicin (Vitrocell). Schistosomula were maintained in 96-well or 24-well culture plates, depending on the experiment, with 200 μL or 1 mL of culture medium M169 (Gibco), respectively, supplemented with 2% fetal bovine serum (Vitrocell), 1 μM serotonin, 0.5 μM hypoxanthine, 1 μM hydrocortisone, 0.2 μM triiodothyronine, penicillin/streptomycin, amphotericin and gentamycin (Vitrocell). Parasites were maintained at 37˚C in 5% $CO_2$ with a humid atmosphere. The medium containing the LSD1 inhibitors or DMSO (vehicle) was refreshed every 24 h during the treatment period (1–4 days).

### Viability assay

An inverted stereomicroscope (Leica M80) was used to evaluate the physiology and behavior of the parasites. Parasites were observed every 24 h, and representative images and videos were

acquired. Schistosomula motility, light opacity, and membrane integrity were evaluated. Adult worm motility, pairing state, adherence to dish surface, and egg laying were monitored and determined. The viability was determined using the CellTiter-Glo Luminescent Cell Viability Assay (Promega) [32]. One thousand schistosomula or 10 paired adult worms´ total cell lysate were submitted to ATP dosage. Eggs laid on the plates were quantified daily and plotted as the ratio of eggs laids per day normalized by the number of coupled worms observed. The compound concentration required to cause mortality in 50% of schistosomula in culture was determined using methods previously described [11]. Briefly, 2000 schistosomula were incubated with different concentrations of MC3935. After 24 h, Alamar Blue reagent (Invitrogen) was added according to the manufacturer's instructions, and the mixture was incubated again for 21 h before samples were analyzed. Detection of viable parasites, which convert the resazurine compound of the reagent into the highly fluorescent resorufin, was performed using a FLUOstar OPTIMA microplate reader (BMG Labtec, Ortenberg, Germany) with the following filter set: Ex 560 nm/Em 590 nm. All of the measurements were performed in triplicate.

## SmLSD1 mRNA quantification

After treatment, the total RNA was extracted using a RiboPure kit (Ambion) followed by DNase treatment (Ambion) and cDNA synthesis (Superscript III, Invitrogen), following the manufacturer's instructions. The resulting cDNA was diluted 10-fold in water and qPCR amplification was performed with 5 μL of diluted cDNA in a total volume of 15 μL using SYBR Green Master Mix (Life Technologies) and specific primer pairs listed in S12 Table were designed for *S. mansoni* genes by Primer3 online software (http://www.bioinformatics.nl/cgi-bin/primer3plus/primer3plus.cgi). QuantStudio 3 Real-Time PCR System (Applied Biosystems) was used. The results were analyzed by the comparative Ct method, the graphs were plotted using GraphPad Prism 8 software, and statistical significance was calculated with the Student´s t-test.

## Western blotting

Nuclear protein extracts from schistosomula or adult worms were prepared as previously described [31]. From each sample, 10 μg of each extract was loaded on 7–12% precast SDS-polyacrylamide gels (Bio-Rad). After transference, membranes were blocked with Tris-buffered saline (TBS) containing 0.1% Tween 20 and 2% bovine serum albumin (TBST/2% BSA) and then probed overnight with specific antibodies in TBS/2% BSA. Membranes were washed with TBST and incubated for 1 h in TBST/2% BSA with secondary antibody (Immunopure goat anti-mouse # 31430, Thermo Scientific, and peroxidase-labeled affinity anti-rabbit # 04-15-06, KPL). After washing the membranes in TBST, the bands were visualized and images were recorded with the Amersham Imaging System (GE Healthcare), and quantified with ImageJ software (NIH). Histone monoclonal antibodies used were anti-H3K4me1 (#5326, Cell Signaling), anti-H3K4me2 (#9725, Cell Signaling) and anti-H3K4me3 (#9727, Cell Signaling), following the manufacturer's instructions. For all antibodies, a 1:1000 dilution was used. For normalization of the signals across the samples, anti-histone H3 antibody (#14269, Cell Signaling) was used.

## Caspase 3/7 activity

The activity of caspase 3/7 was measured using the Caspase-Glo 3/7 assay kit (Promega) following the instructions. Cell lysates from schistosomula were obtained from 2,000 parasites cultivated in a 24-well plate with complete medium (as described above) and treated with MC3935 25 μM or vehicle (DMSO 0.25%). The luminescence was measured in a white-walled

96-well plate in a Wallac Victor2 1420 multilabel counter (PerkinElmer). The graphs were plotted using GraphPad Prism 8 software, and statistical significance was calculated with the Student´s t-test.

## TUNEL assay

Detection of DNA strand breaks in MC3935-treated schistosomula was performed using the *In situ* Cell Death Detection kit (Roche), as previously described [33]. Schistosomula were fixed after 72 h treatment with MC3935 or DMSO. Parasites were mounted in a superfrost glass slide using Prolong with DAPI (Invitrogen) for nuclear visualization. Images were taken on a Zeiss Axio Observer Z1 (Zeiss) inverted microscope equipped with a 40X objective lens and an AxioCam MRm camera in ApoTome mode.

## Confocal laser scanning microscopy

For the confocal microscopy analysis, the adult worms were fixed and stained as previously described [34]. Confocal scanning laser microscopy was performed on a Zeiss LSM 800 microscope equipped with a 488 nm HE/Ne laser and a 470 nm long-pass filter but without the reflection mode. Each slide was analyzed by two independent examiners, and 20 samples were documented to describe the overall observed phenotype.

## Scanning and transmission electron microscopy

Scanning electron microscopy (SEM) and transmission electron microscopy (TEM) were performed to analyze ultrastructural alterations in the parasites. Adult worms or schistosomula were incubated with 25 μM MC3935 or 0.25% DMSO for 48 or 72h, and fixed, as previously described [34]. For SEM analysis, the samples were dehydrated with increasing concentrations of ethanol and then dried with liquid $CO_2$ in a critical-point dryer machine (Leica EM CPD030, Leica Microsystems, Illinois, USA) [35]. Treated specimens were mounted on aluminum microscopy stubs and coated with gold particles using an ion-sputtering apparatus (Leica EM SCD050, Leica Microsystems). Specimens were then observed and photographed using an electron microscope (FEI QUANTA 250, Thermo Fisher Scientific). TEM analysis was performed on a Tecnai G2 microscope (FEI Company). Fixed specimens were washed in 0.1 M cacodylate buffer, pH 7.2; postfixed in 1% $OsO_4$ and 0.8% $K_3Fe (CN)_6$; washed in 0.1 M cacodylate buffer, pH 7.2; dehydrated in a graded acetone series (20˚–100˚ GL) for one hour each step and embedded in Polybed 812 epoxide resin. Ultra-thin sections (60 nm) were collected on copper grids and stained for 30 minutes in 5% aqueous uranyl acetate and for 5 minutes in lead citrate. Each grid was analyzed by two independent examiners, and 20 samples were documented to described the overall observed phenotype.

## Double stranded RNA interference (RNAi)

The coding sequence of lysine-specific histone demethylase 1 (SmLSD1) (GenBank accession #: XM_018797592.1) was amplified by PCR using the oligonucleotides listed in S12 Table, with adult parasite cDNA synthesized using 5 ng of total RNA as template. Two different amplicons (SmLSD1_1 and SmLSD1_2) were used in a second PCR (nested PCR), diluted 1:500, with the oligonucleotides containing an upstream T7 tail sequence (primers are listed in S12 Table). The *GFP* gene was used as a nonrelated dsRNA control and was amplified from pEGFP-N3.

Double-stranded RNA (dsRNA) was synthesized from templates of amplified PCR with oligonucleotides containing the T7 tail. The dsRNA was delivered by soaking the parasite couples in media containing 30 μg/mL of the desired dsRNA, and everyday, 70% of the medium was

changed to a fresh medium also containing 30 μg/mL of dsRNA. At the end of the $2^{nd}$, $4^{th}$, and $7^{th}$ days of incubation, parasites were collected, washed twice in PBS and stored in RNAlater (Ambion) until RNA extraction. At the end of the $2^{nd}$, $4^{th}$ and $7^{th}$ days of incubation the total number of eggs, the number of parasites attached to the plate and the number of couples still paired were quantified. For the H3K4me1 and H3K4me2 western blotting, parasites were collected on the $7^{th}$ day of incubation with dsRNA and stored in PBS at -80°C. The viability of the parasites was determined on the $7^{th}$ day of incubation as described above. Male and female adult worms were ground with glass beads in liquid nitrogen for 5 minutes. RNA extraction and cDNA synthesis were performed as described above. qPCR results were analyzed by the comparative Ct method. Real-time qPCR data were normalized in relation to the level of expression of the Smp_090920 and Smp_062630 reference genes. The graphs were plotted using GraphPad Prism 8 software, and statistical significance was calculated with the Student´s t-test.

### RNA-Seq data analysis

The general outline of the bioinformatics pipeline used for the analysis of the RNA-Seq data is completely described in Pereira et al. [36], including the three different statistical approaches that were used to obtain lists of differentially expressed genes, and considering as the final set only those genes that were listed at the intersection of the three sets. We used the same versions of genome and transcriptome annotation, including the use of a metagenes transcriptome to deal with isoforms, as previously described [36]. All software parameters were as described [36] except for Trimmomatic [37] HEADCROP 12 and MAXINFO 60, since we decided to prioritize longer reads. Each replicate sample of adult worms has generated from 26 to 39 million paired-end 150-bp reads; for schistosomula, a total of 30 to 40 million reads was obtained per replicate sample. The raw RNA-Seq data was deposited at NCBI under BioProject Accession number PRJNA361136.

### Validation of differential gene expression by qRT-PCR

The selection of the candidates was performed based on the higher differences found in gene expression between DMSO and MC3935-treated samples. Only genes found up- or down-regulated in both male and female were selected. The reverse transcription (RT) reaction was performed with 100 ng of each total RNA sample using the SuperScript IV First-Strand Synthesis System (Life Technologies) and random hexamer primers in a 20 μL final volume. The obtained complementary DNAs (cDNAs) were diluted 10 times in DEPC water and quantitative PCR was performed using 2.5 μL of each diluted cDNA in a total volume of 10 μL containing 1X LightCycler 480 SYBR Green I Master Mix (Roche Diagnostics) and 800 nM of each primer in a LightCycler 480 System (Roche Diagnostics). Primers for all transcripts were designed using the IDT Real time PCR tool (https://www.idtdna.com/Primerquest/Home/Index). The sequences of the primers are listed in S12 Table. Each real-time qPCR was run in three technical replicates. The results were analyzed by comparative Ct method (38). Analysis of Relative Gene Expression Data Using Real-Time Quantitative PCR and the 2−ΔΔCT Method. Real-time data were normalized in relation to the level of expression of the Smp_092920 and Smp_062630 as reference genes. The graphs were plotted using GraphPad Prism 8 software, and statistical significance was calculated with the Student´s t-test.

## Results

### *Schistosoma mansoni* lysine specific-demethylase 1

The *Schistosoma mansoni* LSD1 (SmLSD1) contains all three canonical structural, and functional domains (SWIRM, amino-oxidase-like and TOWER domains) (Fig 1A) found in the

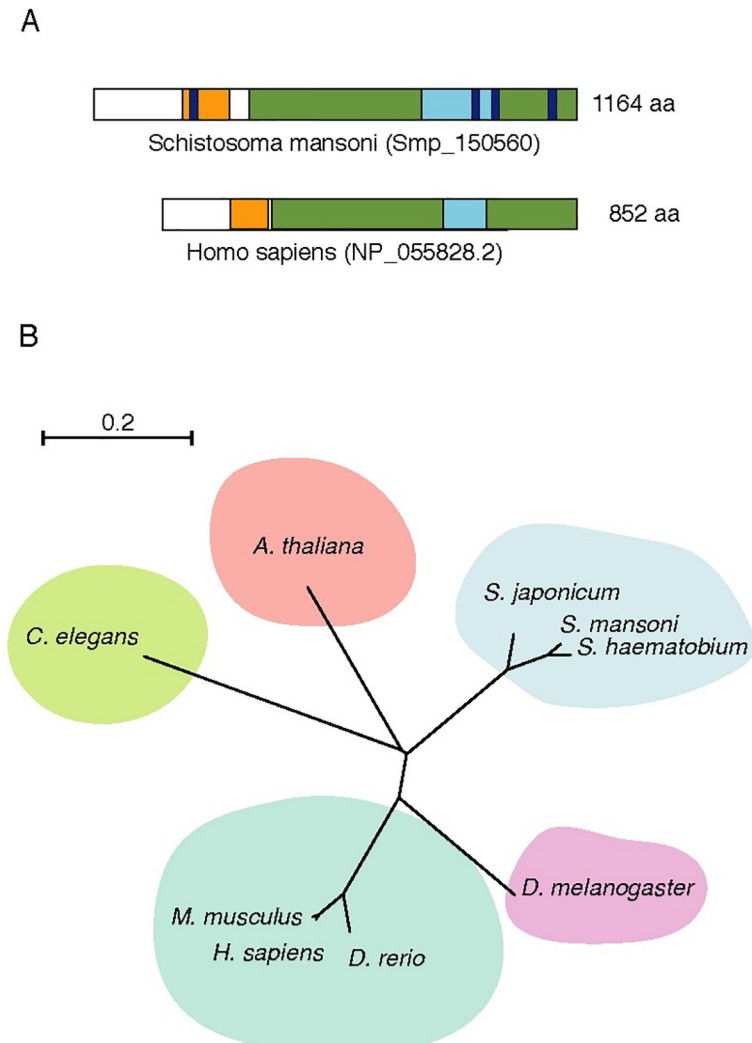

**Fig 1. Overview of SmLSD1 protein domains and conservation.** (A). Schematic representation of the full-length SmLSD1 protein (Smp_150560, top scheme), depicting the conserved functional domains: the SWIRM domain (orange), amine oxidase-like domain (green), the TOWER domain (blue), and schistosome unique sequences (purple). The white rectangles depict a less conserved N-terminal domain (15,5% similarity) as well as a highly conserved hinge region (58,6%) between the SWIRM and amine oxidase-like domains. The full-length of the human LSD1 protein (bottom scheme) is also shown for comparison. (B). Unrooted phylogenetic tree representation was made using the ClustalW2 program and visualized with https://itol.embl.de/.https://itol.embl.de/. Tree scale: 0.2.

LSD1 protein family. We examined in detail the protein alignment between SmLSD1 and human LSD1 (hLSD1) (S2 Fig), particularly since the latter is a well-defined drug target [18] and carried out a limited phylogenetic study including further orthologs. In this regard, our phylogenetic tree revealed that the SmLSD1 protein was closer to human LSD1 than to plant or nematode LSD1 (Fig 1B). It is worth noting that among the five LSD1 homologs tested only SmLSD1 presented unique sequences within all LSD1 functional domains (Fig 1A, purple segments and S2 Fig, dashed lines), which could be explored to develop a *S. mansoni*-selective drug.

## Viability of *Schistosoma mansoni* after MC3935 treatment

Schistosomula were obtained by mechanical transformation of cercariae, and adult worms were recovered by perfusion of infected hamsters (Fig 2A). We screened a series of LSD1 inhibitors (all these small compounds were synthesized based on the scalffold of tranylcypromine (TCP), a well-tested irreversible LSD1 inhibitor) to evaluate their schistosomicidal activity. All compounds at a final concentration of 25 μM showed toxicity against the juvenile form of schistosomula at 72 h (S3A Fig) or adult worms at 96 h of cultivation (S3B Fig). Interestingly, TCP showed the least toxic activity, whereas MC3935 was the most potent compound (S3A and S3B Fig). Therefore, MC3935 was chosen for all further analyses in this study. We showed that MC3935 was able to inhibit the catalytic activity of the recombinant human LSD1 protein (S1B Fig), revealing a 1,000 fold higher inhibitory activity than TCP, proving it as a *bona fide* LSD1 inhibitor. The toxic effect of MC3935 on schistosomula or adult worm pairs was further confirmed (Fig 2B and 2C and S3C Fig). A significant loss of viability at 25 μM MC3935 was observed in schistosomula or adult worms after cultivation for 72 or 96 h, respectively (S3C Fig). These results were confirmed with videos (S1–S4 Videos), which showed nearly 100% of the schistosomula had a complete lack of motility, high granularity and altered body shape (S2 Video) when compared to healthy schistosomula that were treated with DMSO only (S1 Video). MC3935-treated-adult worms also showed significant alterations when compared to the control (Fig 2D and S3 Video), which included unpairing, lack of adherence, extremely low motility, vitellaria involution and no egg laying (Fig 2D and S4 Video). In order to evaluate whether the lack of eggs (Fig 2D; Egg laying) was exclusively due to the separation of the worms (Fig 2D; Pairing), we performed an additional experiment in which only adult worms that were kept coupled were maintained in culture, followed by egg counting. Worm pairs that were not treated, or treated with DMSO laid a significant number of eggs, while worm pairs that received the treatment of MC3935 laid no eggs whatsoever (S4A and S4B Fig).

## Molecular docking and catalytic inhibition of SmLSD1 by MC3935

Since no crystal structure of SmLSD1 is yet available, a homology model of the parasite's enzyme was first generated using an available crystal structure of the orthologous human LSD1. By analyzing the reported crystal structures of hLSD1 in complex with covalently bound tranylcypromine (TCP) derivatives, two crystal structures were found to be most relevant: a crystal structure of the N5 adduct of the highly analogous MC2584, which only lacked the ethynyl group found in MC3935, and the crystal structure with the N5 adduct of the bulky tranylcypromine derivative MC2580 (PDB IDs 2XAQ and 2XAS; respectively [29]). The latter crystal structure (PDB ID 2XAS) was preferred for the use as a template for the homology model since the conformation of residues Glu682 and Asp669 resulted in a more open binding site. Sequence alignment of SmLSD1 and hLSD1 showed an overall sequence identity of 44.1%, while the binding site of the FAD-ligand adduct shared an 80.4% sequence identity. In order to predict the binding mode of MC3935 to SmLSD1, the N5 adduct of this tranylcypromine derivative with the flavin ring of FAD was first generated similar to the N5 adduct of the analogous MC2584, and docking was subsequently performed into the homology model of SmLSD1. The obtained docking pose showed that the N5 adduct of MC3935 adopted a similar orientation in the binding site as observed with MC2584 (Fig 3A) with the ethynyl group embedded between Glu682 and Asp669. Notably, the binding site of SmLSD1 accommodating the tranylcypromine part of the adduct shared a 100% homology with the hLSD1 counterpart (Fig 3B).

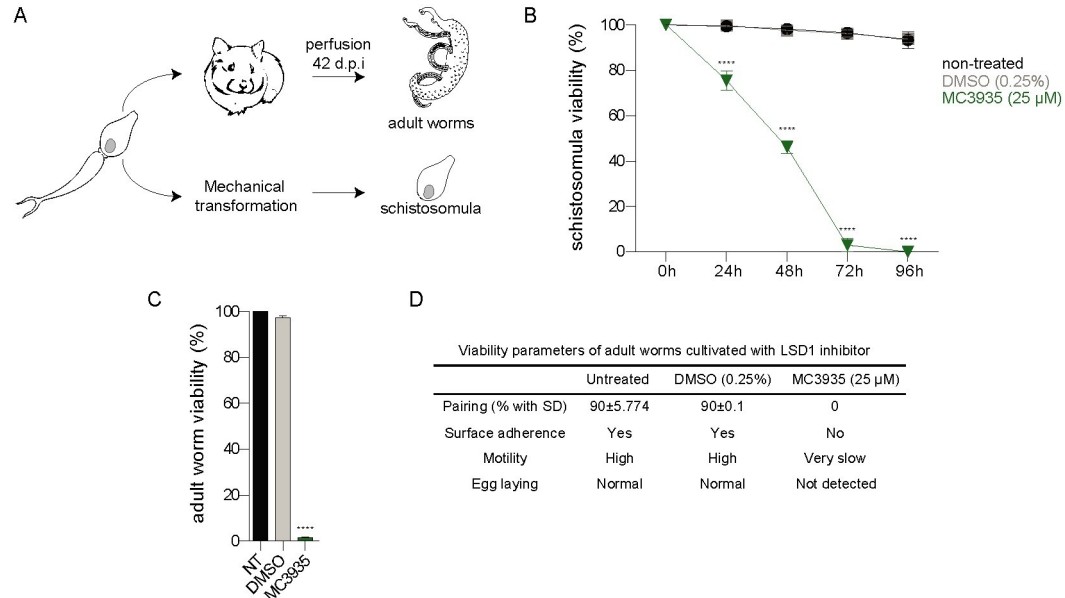

**Fig 2. LSD1 inhibition is detrimental to *Schistosoma mansoni* survival.** (A). Simplified scheme of the acquisition of the two developmental stages of the parasite used in this study. Cercariae were harvested from infected snails and used to either infect hamsters or mechanically transformed into schistosomula for *in vitro* culture. Hamsters were perfused 42–49 days postinfection to harvest adult worm pairs. (B). The relative ATP dosage (%) of schistosomula treated with 25 μM MC3935 (green line) was measured every 24 h (up to 96 h). Schistosomula given DMSO or nothing are shown in gray and black lines, respectively. (C) Relative ATP dosage (%) of adult worm pairs treated with 25 μM MC3935 for 96 h. NT, nontreated adult worm pairs. The results of three independent experiments are shown, and error bars represent the standard deviation (SD). (D) Evaluation of the viability of adult worm treated with DMSO or 25 μM MC3935 for 96 h. Several parameters of adult worm viability were monitored daily until day 4, using an optical microscope equipped with a digital camera. Details for these classifications are described in the methods section. These viability parameters were reviewed and scored by two independent observers. Videos of the control or MC3935-treated worms to confirm the described scores are available (in Supplementary videos). Statistical significance, comparing MC3935-treated and vehicle conditions, was determined using Student´s t-test, with ****p<0.0001.

We next performed western blot analysis and showed that schistosomula or adult worms treated with MC3935 displayed higher band intensities of H3K4me1 or H3K4me2 marks when compared to the DMSO controls (Fig 3C), confirming the inhibition of SmLSD1 demethylase activity. Of note, the increase in H3K4me1 or H3K4me2 methylation was not due to the downregulation of SmLSD1 transcription (S5 Fig, qPCR graphs in A and B). Together, these data confirm that MC3935 inhibited SmLSD1 demethylase activity. Importantly, MC3935 treatments did not alter the H3K4me3 mark in schistosomula or adult worms (S5 Fig, western blots in S5A and S5B), pointing to a selective inhibition of LSD1-specific histone marks.

## Apoptosis in *S. mansoni* after SmLSD1 inhibition

The treatment of schistosomula with MC3935 significantly induced apoptosis, as detected by the 8-fold increase in the activities of caspases 3 and 7 (Fig 4A). In addition, the TUNEL assay indicated extensive double-strand DNA breaks (Fig 4B), as revealed by the green staining of the whole body of the worm treated with the inhibitor. Worms treated with DMSO showed no apoptotic activity, whatsoever (Fig 4A and 4B). These results are in agreement with our data from the ATP viability assay (Fig 2B, S3C Fig) and our observations taken from the optical microscope (S1 and S2 Videos).

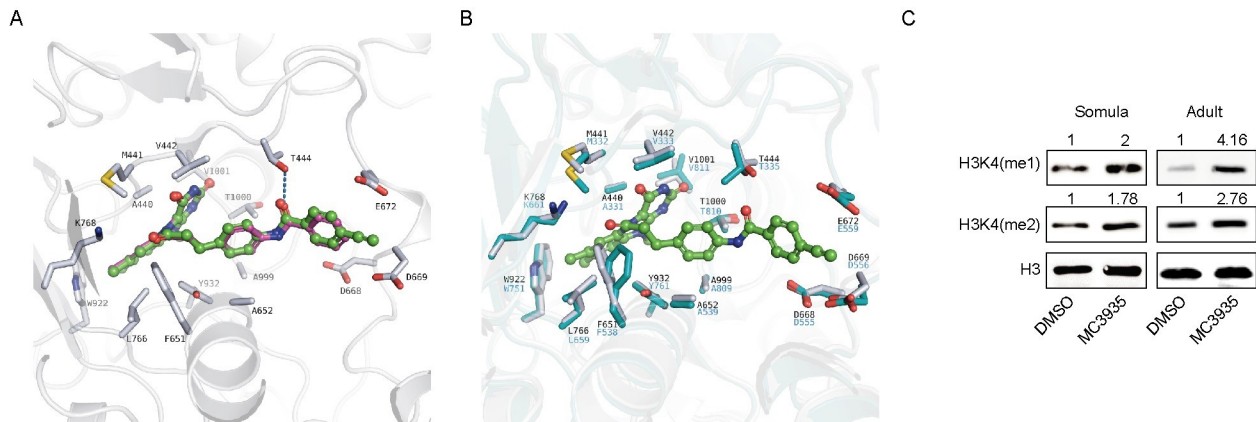

**Fig 3. MC3935 binds to the catalytic pocket of SmLSD1 and inhibits its demethylase activity.** (A). *In silico* molecular docking pose of the N5 adduct of MC3935 (in green) in the homology model of SmLSD1. The experimentally determined binding mode of the analogous MC2584 obtained by the superposition with the corresponding hLSD1 crystal structure (PD ID 2XAQ) is shown in purple. Only side chains of the SmLSD1 binding site are shown (white sticks). (B). Overlay of the SmLSD1 homology model (white sticks) showing the predicted binding mode of the MC3935 adduct with hLSD1 (cyan sticks; PDB ID 2XAS). (C) Western blot of total protein extracts from 72-hour-treated schistosomula (left panels) or 96 hour-treated adult worm pairs (right panels). Monoclonal antibodies against H3K4me1, H3K4me2 and H3 (as loading control) were used. Quantification of the bands (shown above each image) was done by densitometry (ImageJ, NIH software) normalized by the intensity of the H3 band. Western blots were performed from 5 independent biological replicates and one representative is shown here.

## Tegumental damage and ultrastructural abnormalities of schistosomula after SmLSD1 inhibition

The results from our scanning electron microscopy clearly showed that inhibition of SmLSD1 by MC3935 induced severe erosions and fissures in the tegument of schistosomula (Fig 5B and 5D). Worms treated with DMSO revealed the typical healthy status of schistosomula (panel A), showing preserved tegumental spines (panel C). These images corroborate our conclusions

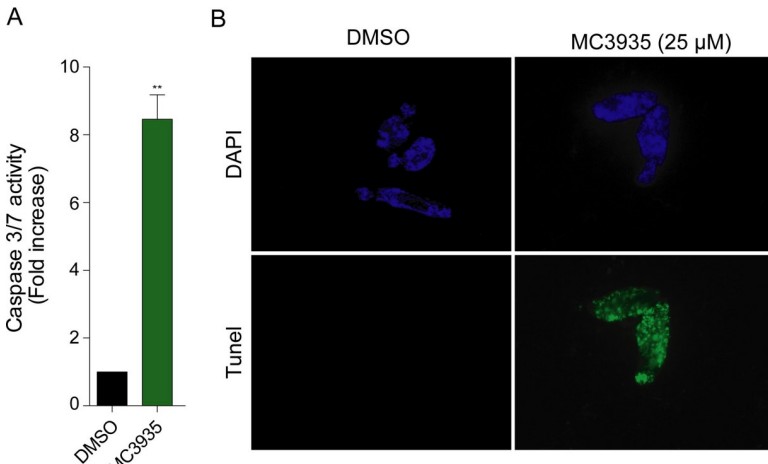

**Fig 4. Inhibition of SmLSD1 triggers apoptosis.** Twenty thousand schistosomula were cultivated in the presence of DMSO or 25 μM MC3935 for 72 h. (A) Caspase 3/7 activity was significantly increased in MC3935-treated parasites (green bar). Statistical significance, comparing MC3935-treated and vehicle conditions, was determined using Student´s t-test, with **p<0.01. (B) TUNEL assay of schistosomula incubated for 72 h with DMSO (left panels) or 25 μM MC3935 (right panels). Green parasites (lower panel) indicate double-strand DNA breaks. DAPI stains nuclear DNA, seen in blue (top panels). Scale bar: 50 μm. The images displayed are representative of three independent experiments, with approximately 20 samples analyzed.

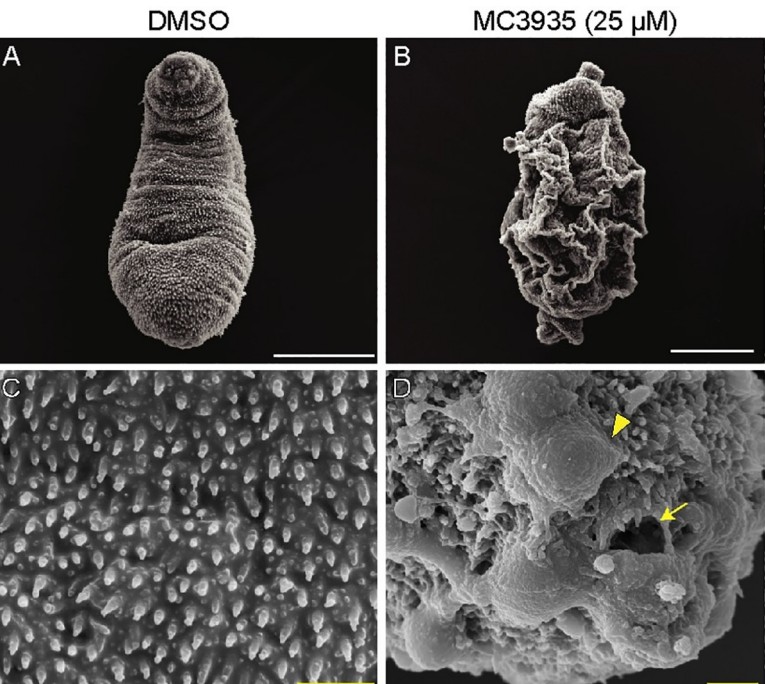

**Fig 5. Inhibition of SmLSD1 leads to tegumental damage of schistosomula.** Schistosomula were treated with 0.25% DMSO (left columns) or 25 μM MC3935 (right columns) for 48 h. Scanning electron microscopy (SEM) images of the tegument in lower (A and B) and higher magnification (C and D). Severe tegumental erosions (arrowhead), and fissures (arrow) are seen at higher magnification (D). Scale bars: white (5 μm) and yellow (1 μm). The images displayed are representative of three independent experiments, with approximately 20 samples analyzed.

that the MC3935 treatment led to schistosomula death. In this respect, it is reasonable to assume that the survival of these worms (note in panels B and D the depth of tegumental damage) could not be rescued even by the eventual withdrawal of the inhibitor.

Transmission electron microscopy of DMSO-treated schistosomula revealed preserved ultrastructures in the worms (Fig 6, panels A, C, E, G and I), such as tegumental spines (black arrows), outer tegument (*), tegument basal lamina (b), circular muscle (cm), longitudinal muscle (lm), mitochondria (m) and nuclei (n). In MC3935-treated schistosomula, extensive ultrastructural disorganization of the tegument was seen, such as a lack of the outer tegument (Fig 6, panel B, asterisks) and tegumental spines (panel B, black arrow). Significant loss of the muscle layers was also observed (compare lm and cm in panels A, C with panels B, D). Large vacuoles were seen in the more external region of the tegument of MC3935-treated parasites (tv in panel D). Additionally, these internal vacuoles contained what seemed to be cellular debris (panel D, #), which could be an indication of tissue degradation and cell death. Panel F shows significant thickening and higher electron density of the outer tegument, associated with the appearance of projections (white asterisk). Loosening and disorganization of the muscle fibers were also observed (panel F and H, lm, and cm) as were different projections of the spines in the outer tegument (panel H, white asterisks). Control parasites (panel I) showed normal and preserved mitochondria (m), always associated with muscle fibers while MC3935-treated parasites (panel J) showed smaller mitochondria (m) that appeared to have less well-defined cristae, and enveloped by membranous structures, which could be an indication of leftover muscle fibers, and they were close to myelin fibers (mf).

## Phenotypic defects of adult *S. mansoni* after SmLSD1 inhibition

Analysis of the adult male tegument and its oral sucker by scanning electron microscopy showed significant alterations upon MC3935 treatment (Fig 7, right panels), when compared to the control worms (Fig 7, left panels). A detailed inspection of the SEM images revealed extensive damage in the dorsal tegument of the male worms that were treated with the LSD1 inhibitor, with the presence of a large number of blisters (Fig 7B and 7D, yellow arrowheads), as well as fissures and holes in the tubercles (Fig 7B and 7D, yellow arrows). Blisters and fissures were also seen in the male oral sucker (Fig 7F, yellow arrows, and arrowheads). Confocal laser scanning microscopy (CLSM) showed important alterations of the sexual organs of MC3935-treated male or female parasites (Fig 7H and 7J). It is worth noting the deleterious effect of the LSD1 inhibitor in the involution of the ovary, leading to a reduced number of mature or immature oocytes (Fig 7, compare panels G and H). The inhibitor also generated severe disorganization of the testicular lobes, culminating in a significantly reduced number of spermatocytes (Fig 7, compare panels I and J). Since the treatment of MC3935 significantly affected the sexual organs of both male and female worms, one should expect that egg production would be severely compromised. S3 and S4 Videos display the described phenotypic abnormalities, which explain the egg laying impairment in MC3935-treated worms. Indeed, inspection of egg laying by the worms cultivated in the presence of MC3935 revealed an almost complete lack of eggs (S4A and S4B Fig and S3 and S4 Videos; in the videos, note the presence of eggs in DMSO-treated worms and a complete lack of eggs in MC3935-treated parasites) a few hours after the addition of the inhibitor. It is also worth noting the involution of the vitellaria in MC3935-treated females (S4 Video).

## Changes in gene expression profile upon SmLSD1 inhibition

We performed RNA sequencing (RNA-Seq) analysis to evaluate the effect of LSD1 inhibition on global gene transcription in *S. mansoni*. Unsupervised hierarchical clustering analysis of RNA-Seq data depicted the changes in global gene expression profile in males, females, or schistosomula upon treatment with MC3935 (Fig 8). Interestingly, inhibition of SmLSD1 significantly modulated 3608 transcribed genes in male, female or schistosomula, with 1964 being downregulated, and 1644 being upregulated (Fig 8A). The highest modulation of gene expression was observed in male worms, for either up- or downregulation (Fig 8A, purple in the Venn diagram), followed by female worms (Fig 8A, pink in the Venn diagram), and schistosomula (Fig 8A, red in the Venn diagram). Importantly, when we analyzed commonly regulated genes in male, female, or schistosomula, we found 220 and 50 genes down- or upregulated, respectively (Fig 8A). The complete lists of significantly upregulated genes in each female, male or schistosomula (S1–S3 Tables), downregulated genes in each females, males or schistosomula (S4–S6 Tables), down- or upregulated genes in common between females and males (S7 and S8 Tables, respectively) are presented in the supporting information. It is noteworthy that in schistosomula, a smaller number of consensus of differentially expressed genes (DEGs) was detected when compared with adult worms (Fig 8A).

A clear profile of the differential gene expression between control and MC3935-treated parasites is depicted in the heatmap (Fig 8B), which confirmed that the treatment led to a significant change in the regulation of genes in females, males and schistosomula, with many genes being either up- (Fig 8B, red) or downregulated (Fig 8B, blue).

A closer look at the twenty most differentially expressed genes in females, males or schistosomula revealed that inhibition of SmLSD1 by MC3935 changed the levels of expression of genes belonging to different critical biological processes, including protein and lipid degradation, RNA processing, calcium and sodium homeostasis, antioxidants and transcription factors

DMSO

MC3935 (25 µM)

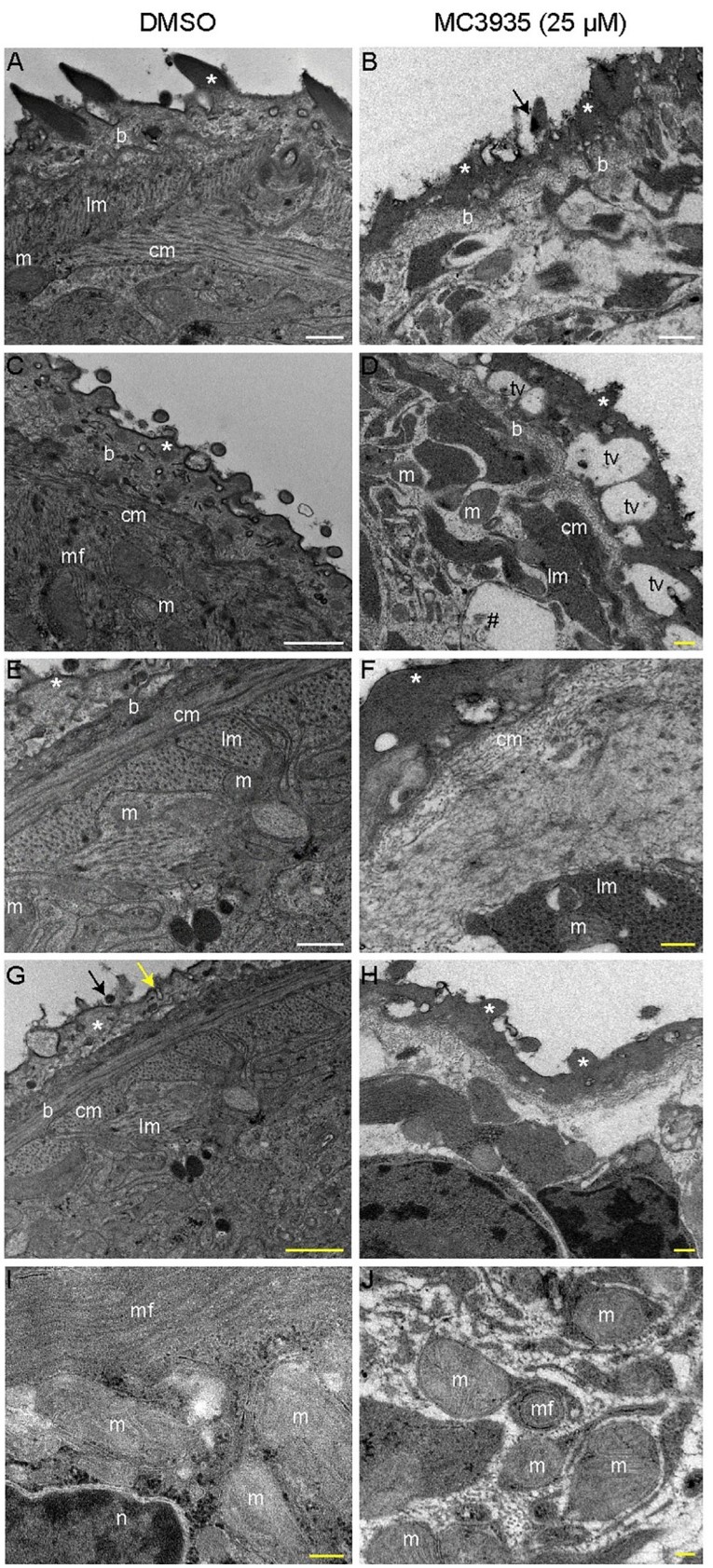

**Fig 6. Inhibition of SmLSD1 leads to ultrastructural abnormalities in schistosomula.** Transmission electron microscopy (TEM) of schistosomula treated with 0.25% DMSO (left column, panels A, C, E, G and I) or 25 μM MC3935 (right column, panels B, D, F, H and J) for 48 h. Symbols are as follows: tegumental spines (black arrows), outer tegument (*), tegument basal lamina (b), circular muscle (cm), longitudinal muscle (lm), mitochondria (m) and nucleus (n). In MC3935-treated schistosomula, an ultrastructural disorganization of the tegument is seen, lacking the outer tegument and tegumental spines (panel B, black arrow). A complete lack of the muscle layers (lm or cm) is also noted (panel B). Large vacuoles are observed in the more external region of the tegument of MC3935-treated parasites (panel D, tv). Internal vacuoles contain cellular debris (panel D, #). Significant thickening and higher electron density of the outer tegument is present, associated with the appearance of projections (panel F, white asterisk). Loosening and disorganization of the muscle fibers (panel F and H, lm and cm) and uncommon projections of the spines in the outer tegument (panel H, white asterisks) are observed. Control parasites (panel I) show normal and preserved mitochondria (m), always associated with muscle fibers (lm or cm). MC3935-treated parasites (panel J) lack muscle fibers, and they show smaller mitochondria (m) that appear to have less defined cristae, to be enveloped by membranous structures and to be close to myelin fibers (mf). Scale bars: white (5 μm) and yellow (1 μm). The images displayed are representative of three independent experiments, with approximately 20 samples analyzed.

(S9–S11 Tables). Within this list, genes encoding proteases stood out as being downregulated in MC3935-treated schistosomula, males and females as compared to DMSO-treated parasites (S9–S11 Tables, shaded in green). Genes encoding protease inhibitors were upregulated in females and males (S9 and S10 Tables, shaded in dark green). Importantly, genes of the digestive system of *S. mansoni* were found down- or upregulated (S9–S11 Tables, shaded in green with *).

Genes encoding kinases and phosphatases were upregulated in treated females and males (S9 and S10 Tables, shaded pink and yellow, respectively). Overall, the analysis of the twenty most differentially expressed genes in treated males revealed a more heterogeneous gene expression profile (S10 Table). Of note, in MC3935-treated schistosomula, a significant number of genes encoding proteins involved in RNA metabolism were upregulated (S11 Table, shaded in blue). Downregulated genes encoding proteases stood out in treated schistosomula (S11 Table, shaded in green).

For gene expression validation, we performed qRT-PCR on the samples submitted to RNA sequencing as well as on cDNA samples from DMSO- or MC3935-treated adult worms (S6 Fig). We selected a set of twelve genes (6 up- and 6 down-regulated) out of the twenty most differentially expressed genes in females and males (S9 and S10 Tables). Importantly, 100% of the genes selected from our RNA-Seq analysis were validated with high significance (S6A and S6B Fig, TPM values). These same twelve genes were used for validation on cDNAs from samples treated under the same conditions of the RNA-Seq experiment (25 μM of MC3935 for 48 h). Again, nearly all the genes were validated for differential expression, with the exeption of *Smp*_139160 and *Smp*_166530 that although did not show significance, point to a clear tendency for down-regulation (S6 Fig, Relative expression). Noteworthy, these validations indicate specificity of gene targeting by MC3935, as well as the robustness of our analysis.

## Knockdown of the SmLSD1 gene

We conducted dsRNAi experiments in adult worm pairs and were able to achieve 70% silencing of SmLSD1 transcription at day 7 (S7A and S7B Fig). Western blot analysis of protein extracts from silenced adult worm pairs revealed a 2-fold inhibition of SmLSD1 demethylase activity (S7C Fig). By monitoring the behavior of the worms on a daily basis, we clearly observed progressive harm in whole-worm physiology during SmLSD1 silencing, which culminated in significant unpairing of male and female worms, a decrease of nearly 50% in the number of laid eggs, low motility and compromised surface adherence by the oral sucker of male worms (S7D–S7F Fig). Silencing of SmLSD1 promoted damage to the oral sucker of male worms (S7G Fig). Of note, the silencing of the parasites with the control dsRNAi showed no

DMSO                                    MC3935 (25 μM)

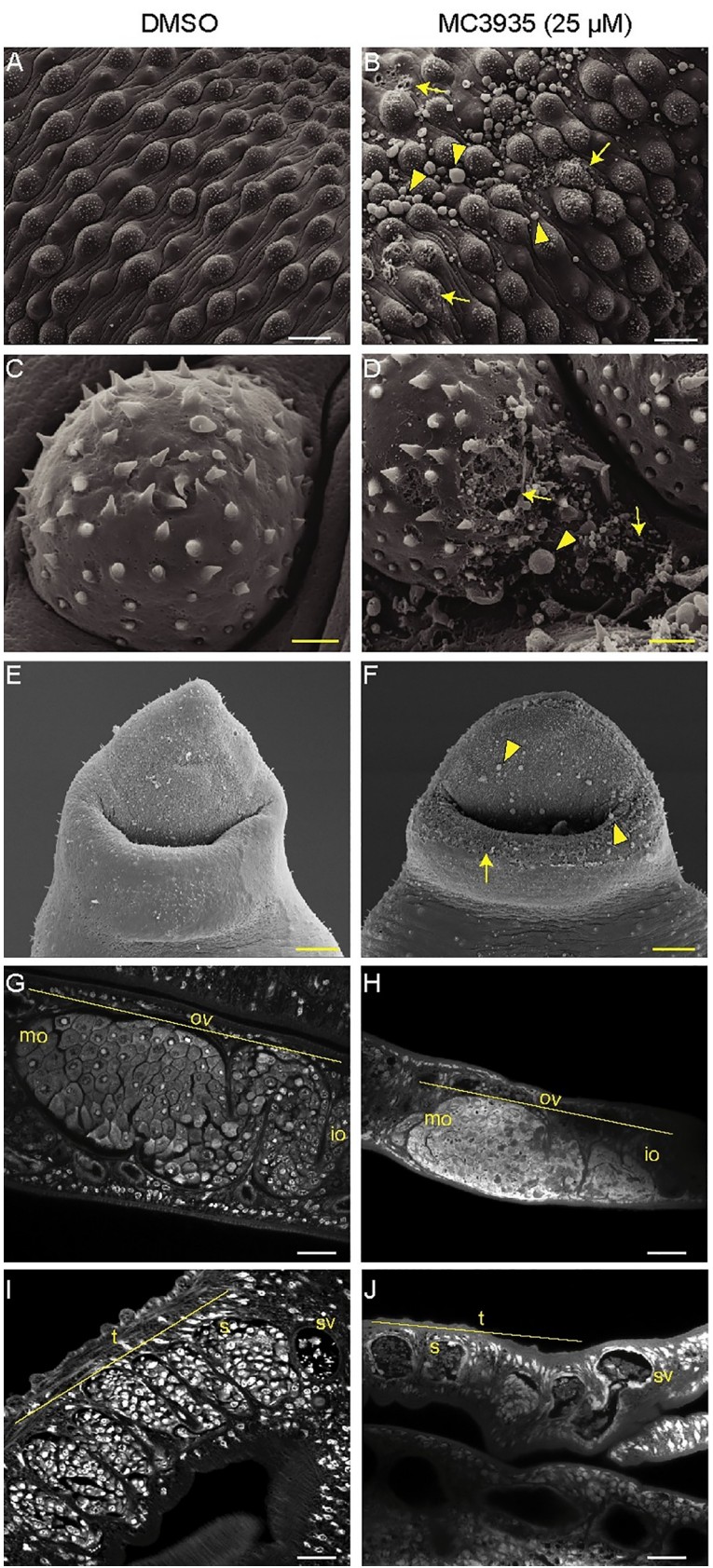

**Fig 7. Inhibition of SmLSD1 leads to tegumental damage and reproductive organ involution in adult schistosomes.** Ten adult worm pairs were cultivated in the presence of DMSO (left column) or 25 μM MC3935 (right column) for 72 h. Scanning electron microscopy (SEM) images from the dorsal region of male worms show damage to the tegument (B), tuberculous (D) and oral sucker (F) compared to controls (A, C and E). Yellow arrows point to fissures and arrowheads point to blisters. Scale bar: 2 μm (yellow). Confocal laser scanning microscopy (CLSM) of ovaries (G and H) and testis (I and J) from control and MC3935-treated, respectively. OV: ovary; mo: mature oocytes; io: imature oocytes; (t) testicular lobes; sv: seminal vesicle; s: spermocytes. Scale bars: 20 μm (white) and 2 μm (yellow). The images displayed are representative of three independent experiments, with approximately 20 samples analyzed.

decrease in SmLSD1 gene expression or activity and no deleterious effects in the worms (S7 Fig, dsGFP).

The twelve differentially expressed genes validated by qRT-PCR from the RNA-Seq analysis (S6A and S6B Fig) were tested in the samples obtained from the RNAi experiment, and we could confirm only one down-regulated gene (*Smp*_156960) (S8 Fig). Although curious, this was not a surprising result, if one considers that there are some caveats involved when dealing with gene silencing by RNA interference, such as partial reduction of the target mRNA levels, and partial reduction in the level of the respective protein due to its possibly low turnover rate. In the case of SmLSD1, we achieved 70% of mRNA silencing, which still leaves 30% of the message to be translated and possibly leaving a considerable amount of LSD1 enzymatic activity for transcriptional regulation. This phenomenon can be easily verified when comparing the levels of mono and dimethylation of histone H3 lysine 4 in the adult worm samples from MC3935 and dsRNA targeting LSD1. We found 4.16-fold increase in H3K4me1 signal in the MC3935-treated worms compared with only a 2.06-fold increase in the dsRNA treated worms. The same was observed in the H3K4me2 signal, where MC3935-treated adult worms presented a 2.76-fold increase in the histone mark whilst dsRNA treated adult worms presented only a 1.84-fold increase in this signal.

Given the fact that the RNA-Seq analysis of the effect of MC3935 has shown that treatment with the drug promotes a broad change in the transcriptional program of the parasite, which affected a number of different metabolic pathways, it is possible that when selecting in a somewhat arbitrary manner the 12 genes that were picked for qPCR validation, we have not hit the set of genes that were most relevant for the phenotypic changes observed by RNAi of SmLSD1.

Nevertheless, SmLSD1 silencing was able to affect the physiology of the parasite (Fig 7E–7G), suggesting that a minimum interference in control of the parasite's transcriptional program is already detrimental to the worms, *in vitro*.

## Discussion

Lysine-specific demethylase 1 (LSD1) is an epigenetic enzyme that oxidatively cleaves methyl groups from mono and dimethyl Lysine 4 of histone H3 (H3K4me1/2) and can contribute to gene silencing [13,38–40]. Since its discovery, LSD1 histone demethylase activity has been investigated as a pharmacologic target for cancer and other diseases. As part of a continuing effort of several different investigators to identify epigenetic modifications as therapeutic targets to control schistosomiasis, a recent paper published during our study also identified *S. mansoni* LSD1 (SmLSD1) as an additional promising drug candidate [17]. In the present study, we provide evidence that SmLSD1 plays important biological roles in the physiology of *S. mansoni* and that inhibition of SmLSD1 by a novel, selective, and potent LSD1 synthetic inhibitor MC3935 is detrimental to the survival of juvenile and adult worms.

MC3935 was synthesized based on the scaffold of the nonselective and irreversible monoamine oxidase (MAO) inhibitor tranylcypromine (TCP) [29,41]. Tranylcypromine is a mechanism-based suicide inhibitor of MAO and LSD1; it covalently binds to the FAD cofactor

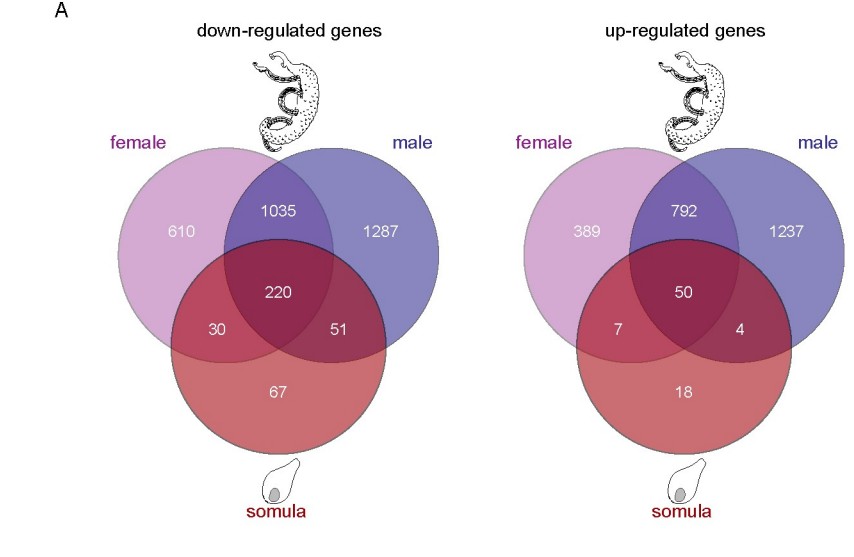

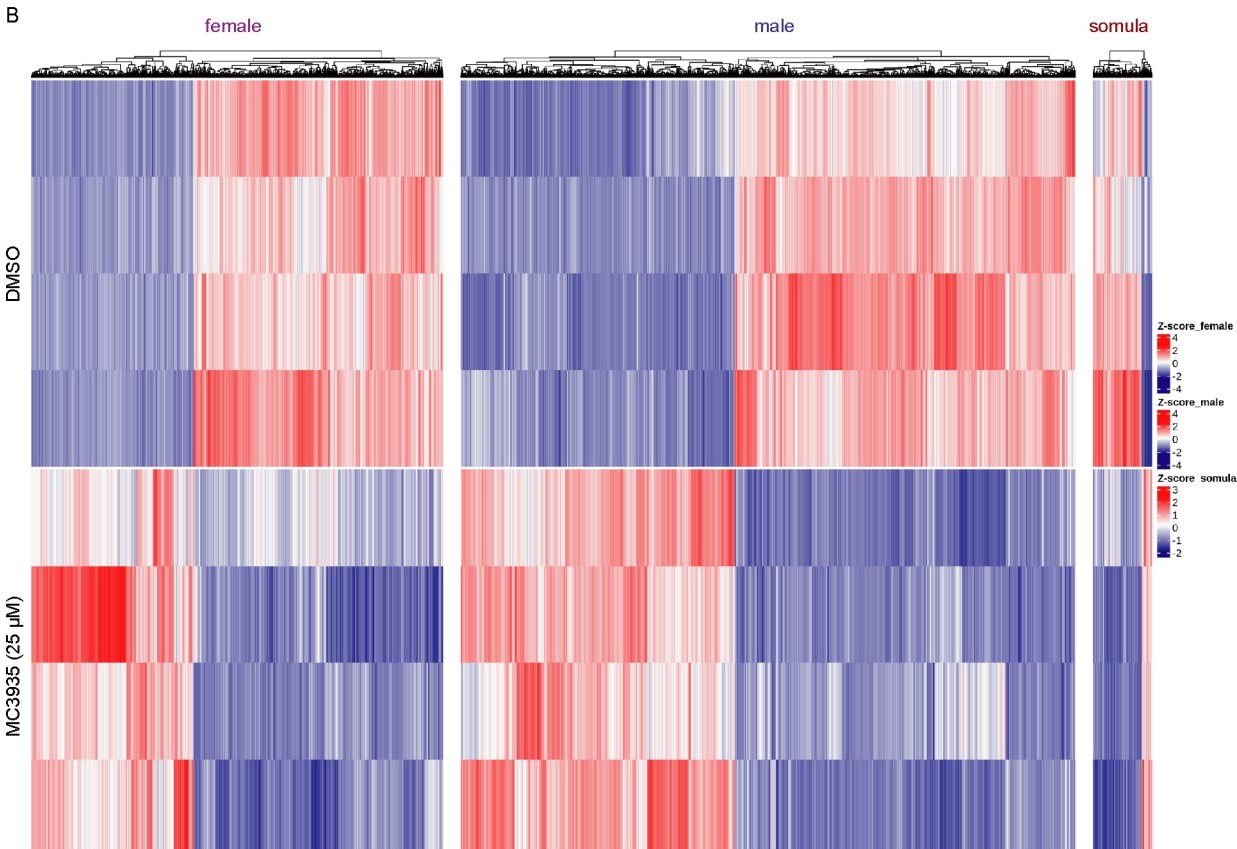

**Fig 8. Inhibition of SmLSD1 triggers genome-wide transcriptional deregulation.** RNA-Seq analysis of female and male worms and schistosomula that were cultivated in the presence of DMSO (control) or MC3935 (25 μM) for 48 hours. (A) Venn diagrams of the number of genes that were detected as differentially expressed among female, male and schistosomula. The numbers at the intersections (darkest red) of the diagrams represent genes commonly affected in the presence of MC3935 among females (purple), males (blue) and schistosomula (red) (220 downregulated genes and 50 upregulated genes). (B) The heatmaps show the hierarchical clustering of differentially expressed genes (columns) in four biological replicates (lines) of female and male worms, and schistosomula, either for controls or for treated parasites, as indicated on the left side of the heatmaps. Blue lines, downregulated genes. Red lines, upregulated genes. Gene expression levels are shown as Z-scores, which represent the number of standard deviations above (red) or below (blue) the mean expression value among treated and control samples for each gene; the expression level Z-scores are color-coded as indicated on the scale at the right side of the heatmap.

embedded within the protein, thus abolishing enzymatic catalysis [42]. Importantly, MC3935 showed a 1,000 fold higher inhibitory activity than TCP, *in vitro*. In addition, our *in silico* molecular docking indicated that MC3935 could indeed be an effective SmLSD1 inhibitor and that the inhibition also relied on the amino-oxidase-like catalytic domain. Toxicity assays on *S. mansoni* parasites using each of our 11 synthetic putative LSD1 inhibitors, as well as TCP, revealed MC3935 as the most powerful, with TCP showing the lowest toxicity toward the worms. This was a surprising result if we consider that both MC3935 and TCP [43] should adopt a similar orientation in their binding to the catalytic site of SmLSD1, based on our *in silico* molecular docking. However, one could always assume that the lack of toxicity of TCP was due to its lower permeability to the tegument of *S. mansoni* or that it is more metabolically labile. Indeed, MC3935 fills better the catalytic tube of the enzyme better than TCP, due to its addiotional ethynylbenzamide portion, which allows additional interaction within the tube. Importantly, our western blot analyses indicated that MC3935 specifically inhibited H3K4 mono- and dimethylation, which are known LSD1 targets, but not H3K4 trimethylation, a mark that is targeted by the histone demethylases of the Jumonji family [44].

Genetic studies in numerous models have suggested that LSD1 plays a significant role in developmental processes [45]. LSD1 has also been reported to have a role in the DNA damage response [46], repression of mitochondrial metabolism, lipid oxidation energy expenditure programs [47,48], and smooth muscle regeneration [49]. Additionally, germline murine knockouts exhibit embryonic lethality before E7.5: the egg cylinder fails to elongate and gastrulate, resulting in development arrest [50]. In *C. elegans*, the homolog spr-5 regulates Notch signaling [51,52], and maintains transgenerational epigenetic memory and fertility [53]. The yeast spLsd1/2 [54] and *Drosophila* Su(var)3-3 [55] homologs regulate gene silencing, which is required to guarantee normal oogenesis [56] and spermatogenesis [57] in *Drosophila*.

Taking into account the phenotypic effects observed in schistosomula or adult worms under the treatment with the LSD1 inhibitor MC3935, it can be assumed that SmLSD1 also plays important roles in the development and homeostasis of the tegument, muscle and sexual organs of *S. mansoni*. Although inhibition of SmLSD1 can have more pronounced effects on many other key biological processes or structures in *S. mansoni*, the tegument disruption would alone represent a desirable LSD1 target. The tegument of *S. mansoni* plays a crucial role in its protection against the host immune system [58]; it is capable of absorbing nutrients and molecules, as well as excreting metabolites and synthesizing proteins [59,60]. Inhibition of SmLSD1 activity, either by irreversible MC3935 binding or by partial knockdown of the SmLSD1 gene, generated pronounced damage in schistosomula or adult worms (Fig 5, Fig 7, and S7 Fig), which likely made a major contribution to the observed mortality of the parasites within a short period of time (Fig 2, Fig 4, S3 Fig, S7 Fig, and supplementary videos).

The musculatory activity is essential in several aspects of *S. mansoni* biology and physiology, including infection, pairing, feeding, regurgitation, reproduction and egg laying [61–63]. Our transmission electron microscopy revealed a complete lack of muscle layers in schistosomula treated with the LSD1 inhibitor (Fig 6), which was accompanied by phenotypic defects observed in treated-worms, such as lack of motility, sucker adherence, egg laying and vitellaria contraction (Fig 2, S4 Fig and S4 Video), that could be related to the compromised muscle structures.

Interestingly, the role of LSD1 in oogenesis and spermatogenesis seems to be conserved among different organisms, including human, *C. elegans*, *Drosophila*, and *S. mansoni* (this paper). Our confocal microscopy revealed significant alterations in the sexual organs of treated worm pairs, such as a reduced number of spermatocytes and oocytes (Fig 7H and 7J). These data are in agreement with the incapacity of the worms to produce eggs, even when remaining paired during the treament (see S4 Fig).

Several diseases have been associated with aberrant histone methylation/demethylation patterns. Thus, considering that juvenile or adult *S. mansoni* are highly transcriptionally active and that SmLSD1 is expected to control the expression of a variety of different genes to maintain the homeostasis of the parasites, we compared their global transcriptional profiles after incubation with a sublethal dose of MC3935. Our RNA-Seq analysis revealed that inhibition of SmLSD1 led to the differential expression of genes involved in several different biological processes. These data suggest that SmLSD1 is recruited to target gene promoters by different transcription factors. Among the genes that were downregulated upon SmLSD1 inhibition in schistosomula, females and males, several proteases stood out (S9–S11 Tables, shaded in green), mainly from the cathepsin family, including two hemoglobinases in female worms. This is an important finding considering that proteases are key components of the pathogenicity of the parasite; they facilitate tissue penetration and determine the nutritional sources of the parasite within intermediate and human hosts [64]. Importantly, two protease inhibitors were upregulated in females and males (S9 and S10 Tables, shaded in dark green). That SmLSD1 controls the expression of genes of the digestive system of female or male worms was supported by the fact that a number of genes encoding enzymes or proteins of the digestive tract of the parasite [65] had modified expression in treated parasites (S9–S11 Tables, shaded in green and with *). In this regard, it is noteworthy that blood digestion seemed to be severely compromised in females (see S4 Video).

Interestingly, we identified the SMDR2 gene as significantly downregulated in female worms treated with the LSD1 inhibitor (S9 Table, shaded in green and with #). SMDR2 is the schistosome homolog of P-glycoprotein (PgP) [66], an ATP-dependent efflux pump, and its downregulation upon treatment might be involved in the high toxicity of MC3935 observed in females (see S4 Video).

Importantly, a significant number of proteins involved in RNA metabolism seemed to be specifically upregulated in schistosomula (S11 Table, shaded in blue). This finding is of great importance since, although histone methylation has been only recently coupled to RNA processing [67], this is the first report suggesting a role for LSD1 in regulating RNA metabolism.

The present study further suggests SmLSD1 is a promising candidate as a drug target. More generally, the epigenetic regulation of chromatin affects many fundamental biological pathways. Therefore, the realization that the deregulation of chromatin is harmful to *S. mansoni* should reinforce significant efforts to develop selective epigenetic drugs against schistosomiasis. Although the results obtained do not allow us to pinpoint a specific mechanism to explain the toxicity of the inhibitor, and we cannot rule out some off-target effects.

## Supporting information

**S1 Fig. Synthesis and IC$_{50}$ of MC3935.** (A). Compound **1** (MC3935) was synthesized by coupling racemic *tert*-butyl (*trans*-2-(4-(4-ethynylbenzamido)phenyl)cyclopropyl) carbamate **2,** prepared as previously reported,[1] with the commercially available 4-ethynylbenzoic acid followed by acidic deprotection of the Boc protected amine **3.** Reagents for the synthesis of compound **1** (**MC3935**): (a) HOBt, EDCI, TEA, dry DMF, rt; (b) HCl 4N in dioxane, dry THF, 0˚C-rt. (B). MC3935 inhibits the catalytic activity of recombinant human LSD1 (hLSD1). The concentration required to inhibit the activity of the purified hLSD1 protein by 50% (IC$_{50}$) is shown in the graph. (C). Schistosomula viability is impaired by MC3935 treatment. The concentration required to cause mortality in 50% of the parasites (IC$_{50}$) after 48 hours is shown in the graph.
(TIF)

**S2 Fig. *Schistosoma mansoni* LSD1 protein alignment.** Sequence alignment using the Clustal Omega tool was performed including the *Homo sapiens*—NP_055828, and *Schistosoma mansoni*–XP_018652619.1. The functional domains of the LSD1 protein family are underlined as follows: the SWIRM domain in orange (165–287 aa), the amino-oxidase-like domain in green (379–824 and 909–1136 aa) and the TOWER domain in blue (825–908 aa). Amino acid positions refer to the SmLSD1 protein. Unique amino acid sequences found within the SmLSD1 polypeptide are shown as dashes.
(TIF)

**S3 Fig. Screening of synthetic small LSD1 inhibitors in *Schistosoma mansoni*.** Twenty thousand schistosomula (A) or ten adult worm pairs (B) were incubated with 25 μM of LSD1 inhibitors or DMSO (nontreated parasites were included as an additional control) and submitted to an ATP cell viability assay. Tranylcypromine (TCP, red bars) is a well-known irreversible LSD1 inhibitor. Twelve different compounds based on the TCP scaffold were tested. (C) Dose-dependent toxicity of MC3935 (at 1, 10 or 25 μM) on schistosomula or adult worm pairs. Incubation times for schistosomula and adult worms were 72 h and 96 h, respectively. The results of three independent assays are shown; error bars represent the SD. Statistical significance, comparing treated and vehicle conditions, was determined using one-way ANOVA, with $*p<0.05$, $**p<0.01$ and $****p<0.0001$, and ns standing for non-significant.
(TIF)

**S4 Fig. LSD1 inhibition affects egg production.** (A). Adult worm pairs were treated (or not, NT) with 0.25% DMSO or 25 μM MC3935 and cultivated for 96 h. The number of laid eggs was counted daily and a representative image was recorded. Scale bar: 250 μm. (B). Quantification of eggs normalized by the number of adult worm pairs and days of treatment. Statistical significance, comparing MC3935-treated and vehicle conditions, was determined using Student´s t-test, with $****p<0.0001$.
(TIF)

**S5 Fig. SmLSD1 inhibition by MC3935 has no effect on H3K4 trimethylation in schistosomes.** Quantitative RT-PCR analysis of SmLSD1 mRNA or western blot analyses of SmLSD1 protein from schistosomula (A) or adult worms (B) after 72 h or 96 h incubation time with MC3935, respectively. The bars indicate standard deviations from three independent measurements. Histone H3 was included in western blots as the loading control. Statistical significance, comparing MC3935-treated and vehicle conditions, was determined using Student´s t-test, with ns standing for non-significant.
(TIF)

**S6 Fig. Validation of gene expression data by quantitative reverse transcriptase-real time PCR.** The expression of twelve selected genes was measured by qRT-PCR in RNA samples extracted from male or female parasites exposed for 48 h *in vitro* either to DMSO (gray bars) or 25 μM of MC3935 (green bars). Top panels in A or B show the validation of down-regulated or up-regulated genes, respectively, from samples submitted to RNA-Sequencing (TPM values). Bottom panels in A or B show the validation of down-regulated or up-regulated genes, respectively, from samples submitted to cDNA synthesis (Relative expression $2^{\wedge}\Delta Ct$). Expression was normalized as indicated in the Methods, and the lowest normalized value among the control biological replicates was chosen as reference and arbitrarily set to 1. Relative expression of all other control and treated samples was calculated in relation to that value. Graphs show the mean +_ S.D. of four biological replicates for each condition in males and females. Statistical significance, comparing MC3935-treated and vehicle conditions, was evaluated with the Student´s t-test and significant changes are marked by asterisk with $*p<0.05$, $**p<0.01$,

***p<0.001 and ****p<0.0001. Down-regulated genes: *Smp*_055780: smdr2; *Smp*_085180: cathepsin B (C01 family); *Smp*_126120: LAMA protein 2; *Smp*_014570: Saposin1; *Smp*_139160: SmCL2 peptidase (C01 family); *Smp*_166530: phospholipase A. Up-regulated genes: *Smp*_034500: Dual specificity protein phosphatase 10; *Smp*_147730: single Kunitz protease inhibitor; *Smp*_025390: putative calcium dependent protein kinase; *Smp*_128550: src type protein tyrosine kinase; *Smp*_094930: early growth response protein 1; *Smp*_047660: ferritin2C heavy polypeptide 1.
(TIF)

**S7 Fig. SmLSD1 knockdown partially recapitulates MC3935 phenotypes in adult worms.**
(A). Adult worm pairs were soaked with 30 μg of dsDNA and cultivated for up to 7 days. A silencing of 70% was obtained for SmLSD1 mRNA at day 7 (panels A and B). (C) Western blot analysis of total protein extracts from GFP- or SmLSD1-silenced worms at day 7. Band intensity quantifications (obtained with Image J) are shown above each image, and they were normalized by the H3 band. (D and E) Egg production by GFP- or SmLSD1-silenced female worms was monitored daily (scale bar = 200 μm). (F) Several parameters for adult worm viability were monitored daily using a light microscope, until day 7. The viability parameters were reviewed and scored by two independent observers. (G) SmLSD1 RNAi-mediated phenotypic effects observed by scanning electron microscopy. Arrows point to fissures and arrowheads to blisters in the oral sucker of male worms (scale bar = white (5 μm) and yellow (1 μm). Statistical significance was determined using one-way ANOVA, with **p<0.01, ***p<0.001 and ****p<0.0001. The images displayed are representative of three independent experiments, with approximately 20 samples analyzed.
(TIF)

**S8 Fig. Validation of gene expression data by quantitative reverse transcriptase-real time PCR.** Panels A and B show the expressions of selected genes that were measured by qRT-PCR. The cDNA samples were from adult *S. mansoni* incubated for seven days with double-stranded RNA interference for the SmLSD1 gene or the negative control GFP gene. The methodology of the qRT-PCR and the twelve selected genes (*Smp*) were the same as described in S7 Fig. Statistical significance was evaluated with the Student's t-test and significant changes are marked by asterisk with *p<0.05.
(TIF)

**S1 Table. List of upregulated genes on MC3935-treated female adult worms.**
(XLSX)

**S2 Table. List of upregulated genes on MC3935-treated male adult worms.**
(XLSX)

**S3 Table. List of upregulated genes on MC3935-treated schistosomula.**
(XLSX)

**S4 Table. List of downregulated genes on MC3935-treated female adult worms.**
(XLSX)

**S5 Table. List of downregulated genes on MC3935-treated male adult worms.**
(XLSX)

**S6 Table. List of downregulated genes on MC3935-treated schistosomula.**
(XLSX)

**S7 Table. List of commonly downregulated genes on MC3935-treated male and female adult worms.**
(XLSX)

**S8 Table. List of commonly upregulated genes on MC3935-treated male and female adult worms.**
(XLSX)

**S9 Table. List of the 20 most differently expressed genes on MC3935-treated female adult worms.**
(DOCX)

**S10 Table. List of the 20 most differently expressed genes on MC3935-treated male adult worms.**
(DOCX)

**S11 Table. List of the 20 most differently expressed genes on MC3935-treated schistosomula.**
(DOCX)

**S12 Table. List of primers.**
(DOCX)

**S1 Video. Schistosomula treated with DMSO for 48 h.**
(AVI)

**S2 Video. Schistosomula treated with MC3935 for 48 h.**
(AVI)

**S3 Video. Adult worms treated with DMSO for 96 h.**
(MP4)

**S4 Video. Adult worms treated with MC3935 for 96 h.**
(MP4)

## Acknowledgments

We thank Mr. Paulo Cesar dos Santos (Fiocruz, Rio de Janeiro, Brazil) for providing *S. mansoni* cercariae. We are in debt to Dr. Geetha Venkatesh for critical comments and valuable discussions on the bioinformatics analysis.

## Author Contributions

**Conceptualization:** Vitor Coutinho Carneiro, Raymond J. Pierce, Marcelo Rosado Fantappié.

**Formal analysis:** Isabel Caetano de Abreu da Silva, Sergio Verjovski-Almeida, Dante Rotili, Antonello Mai, Wolfgang Sippl, A. Ganesan, Marcelo Rosado Fantappié.

**Funding acquisition:** Raymond J. Pierce.

**Investigation:** Vitor Coutinho Carneiro, Murilo Sena Amaral, Adriana S. A. Pereira, Gilbert Oliveira Silveira, David da Silva Pires, Eduardo José Lopes-Torres, Dina Robaa, Amanda Roberta Revoredo Vicentino, Fernanda Sales Coelho.

**Methodology:** Isabel Caetano de Abreu da Silva, Frank J. Dekker, Dante Rotili, Antonello Mai, M. Teresa Borrello, Julien Lancelot.

**Project administration:** Marcelo Rosado Fantappié.

**Resources:** Sergio Verjovski-Almeida, Raymond J. Pierce, Silvana Thiengo, Monica Ammon Fernandez, Marina Moraes Mourão, Marcelo Rosado Fantappié.

**Supervision:** Sergio Verjovski-Almeida, Wolfgang Sippl, A. Ganesan, Marina Moraes Mourão, Marcelo Rosado Fantappié.

**Visualization:** Vitor Coutinho Carneiro.

**Writing – original draft:** Marcelo Rosado Fantappié.

**Writing – review & editing:** Vitor Coutinho Carneiro, Isabel Caetano de Abreu da Silva.

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
