## [Decision Letter · Decision Letter 0]

29 Dec 2019

Dear Dr. Prof. Fantappié:

Thank you very much for submitting your manuscript "Pharmacological inhibition of lysine-specific demethylase 1 (LSD1) induces global transcriptional deregulation and ultrastructural alterations that impair viability in Schistosoma mansoni" (#PNTD-D-19-01913) for review by PLOS Neglected Tropical Diseases. Your manuscript was fully evaluated at the editorial level and by independent peer reviewers. The reviewers appreciated the attention to an important problem, but raised some substantial concerns about the manuscript as it currently stands. These issues must be addressed before we would be willing to consider a revised version of your study. We cannot, of course, promise publication at that time.

We therefore ask you to modify the manuscript according to the review recommendations before we can consider your manuscript for acceptance. Your revisions should address the specific points made by each reviewer. 

When you are ready to resubmit, please be prepared to upload the following:

(1) A letter containing a detailed list of your responses to the review comments and a description of the changes you have made in the manuscript.

(2) Two versions of the manuscript: one with either highlights or tracked changes denoting where the text has been changed (uploaded as a "Revised Article with Changes Highlighted" file); the other a clean version (uploaded as the article file).

(3) If available, a striking still image (a new image if one is available or an existing one from within your manuscript). If your manuscript is accepted for publication, this image may be featured on our website. Images should ideally be high resolution, eye-catching, single panel images; where one is available, please use 'add file' at the time of resubmission and select 'striking image' as the file type. 

Please provide a short caption, including credits, uploaded as a separate "Other" file. If your image is from someone other than yourself, please ensure that the artist has read and agreed to the terms and conditions of the Creative Commons Attribution License at http://journals.plos.org/plosntds/s/content-license (NOTE: we cannot publish copyrighted images). 

(4) If applicable, we encourage you to add a list of accession numbers/ID numbers for genes and proteins mentioned in the text (these should be listed as a paragraph at the end of the manuscript). You can supply accession numbers for any database, so long as the database is publicly accessible and stable. Examples include LocusLink and SwissProt.

(5) To enhance the reproducibility of your results, we recommend that you deposit your laboratory protocols in protocols.io, where a protocol can be assigned its own identifier (DOI) such that it can be cited independently in the future. For instructions see http://journals.plos.org/plosntds/s/submission-guidelines#loc-methods

While revising your submission, please upload your figure files to the Preflight Analysis and Conversion Engine (PACE) digital diagnostic tool, https://pacev2.apexcovantage.com/ PACE helps ensure that figures meet PLOS requirements. To use PACE, you must first register as a user. Then, login and navigate to the UPLOAD tab, where you will find detailed instructions on how to use the tool. If you encounter any issues or have any questions when using PACE, please email us at figures@plos.org.

We hope to receive your revised manuscript by Feb 27 2020 11:59PM. If you anticipate any delay in its return, we ask that you let us know the expected resubmission date by replying to this email.

To submit a revision, go to https://www.editorialmanager.com/pntd/ and log in as an Author. You will see a menu item call Submission Needing Revision. You will find your submission record there. 

Sincerely,

Wei Hu

Guest Editor

Timothy Geary

Deputy Editor

Reviewer's Responses to Questions

**Key Review Criteria Required for Acceptance?**

**Methods**

-Are the objectives of the study clearly articulated with a clear testable hypothesis stated?

-Is the study design appropriate to address the stated objectives?

-Is the population clearly described and appropriate for the hypothesis being tested?

-Is the sample size sufficient to ensure adequate power to address the hypothesis being tested?

-Were correct statistical analysis used to support conclusions?

-Are there concerns about ethical or regulatory requirements being met?

Reviewer #1: The approach to answering the question of the role of LSD1 as a drug target in S. mansoni schistosomula and adult worms was detailed and extensive. All regulatory and ethical requirements were met in my judgement.

Reviewer #2: (No Response)

Reviewer #3: Please add the methods of statistical analysis in the article.

**Results**

-Does the analysis presented match the analysis plan?

-Are the results clearly and completely presented?

-Are the figures (Tables, Images) of sufficient quality for clarity?

Reviewer #1: The results cover an array effects of the inhibitor MC3935 on schistosomula and adult worms . They provide convincing evidence of the affect of the inhibitor on on LSD1 and indicate that LSD1 is an appropriate target for new drug therapy. 

The figures especially 4A,B; 6; 7G,H; 8 all suffer from a lack of clarity of the labeling. All the figures need to be checked. 

Figure 1, the different areas except the white are defined.

Reviewer #2: (No Response)

Reviewer #3: 1.The adult worm motility and egg laying assay should use the paired adult worms, but not mixing with paired or unpaired.

2.How many parasites have been observed in each group in figure 5, 6 and 7? Are all the same in each group? Can you provide the statistical results?

**Conclusions**

-Are the conclusions supported by the data presented?

-Are the limitations of analysis clearly described?

-Do the authors discuss how these data can be helpful to advance our understanding of the topic under study?

-Is public health relevance addressed?

Reviewer #1: Conclusions are well supported by the data

Any limitations are clearly described eg why docking instead of crystal structure

The outcomes of the experiments significantly advance the field and suggest a new important drug target

Reviewer #2: (No Response)

Reviewer #3: (No Response)

**Editorial and Data Presentation Modifications?**

Reviewer #1: The figures especially 4A,B; 6; 7G,H; 8 all suffer from a lack of clarity of the labeling. All the figures need to be checked. 

Figure 1, the different areas except the white are defined. 

Line 606 tube not tuibe

Reviewer #2: (No Response)

Reviewer #3: (No Response)

**Summary and General Comments**

Reviewer #1: This study focusses on a lysine-specific demethylase (SmLSD1) as a potential drug target to treat schistosomiasis. The study employs a specific inhibitor, MC3935. By using viability assays and scanning and ttransmission electron microscopy, the authors show that treatment with MC3935 affected parasite motility, egg-laying, tegument, and cellular organelle structures, culminating in the death of schistosomula and adult worms. In silico molecular modeling and docking analysis indicates that MC3935 binds to the catalytic pocket of SmLSD1. Western blot analysis revealed that MC3935 inhibited SmLSD1 demethylation activity of H3K4me1/2. Knockdown of SmLSD1 by RNAi recapitulated MC3935 phenotypes in adult worms. RNA-seq analysis of MC3935-treated parasites revealed significant differences in gene expression related to critical biological processesThe outcomes of a series of experiments show that SmLSD1 is a promising drug target for the treatment of schistosomiasis affecting both schistosomula and adult worms. The significance of the work is that schistosomiasis control relies on mass drug therapy with praziquantel. However, praziquantel is the only drug in use and evidence for drug resistance has emerged. Therefore, identification of novel drug targets in addition to those of praziquantel represent a significant advance. This paper provides convincing evidence for SmLSD1 as an important drug target that deserves further study. The authors are to be congratulated on a well written, organized and executed study.

Reviewer #2: Schistosomiasis is an important NTD affecting hundreds of millions of people in the developing world and treatment relies on a single drug. Therefore, the discovery of new therapeutics and therapeutic targets is important. Here, Carneiro et al. report the effects of a series of potential inhibitors of the LSD1 histone demethylase on Schistosoma mansoni. They show that many of these potential LSD1 inhibitors at relatively high concentrations (25 micromolar) have profound morphological effects on worms. They go on to show that one of these inhibitors MC3935 causes changes in both histone methylation in worms and alternations in gene expression. Although the authors bring together a number of tools to evaluate the effects of these compounds on the parasites, it remains unclear if these compounds are acting via LSD1 to have specific effects on worms. Therefore, I suggest the following revisions:

1) Do all the compounds evaluated inhibit human (or ideally schistosome) LSD1 activity equally in vitro? Can the authors relate the IC50 for enzyme inhibition to activity against worms? i.e., do the most potent LSD1 inhibitors have the most potent effects on worms and vice versa?

2) The authors report that drug treatment causes specific transcriptional changes, that they attribute to changes in histone methylation. The authors also report the RNAi of LSD1 causes worms to get sick, similar to drug treatment, again attributed to less LSD1 activity. Can the authors show that RNAi of LSD1 causes similar changes in mRNA expression as drug treatment. This could provide some evidence that the effects of drugs are via LSD1 inhibition. RNAseq would be best, but qPCR could also address this point.

Reviewer #3: The article submitted present the results that pharmacological inhibition of lysine-specific demethylase 1 (LSD1) of Schistosoma mansoni induced impairing parasite motility, egg-laying, tegument, and cellular organelle structures, culminating in the death of schistosomula and adult worms.

The information included is of interest and merit to be published after major revisions as follows: 

1.The authors concluded that SmLSD1 is a promising drug target for the treatment of schistosomiasis. However, in the context, these effects of LSD1 inhibitor on S. mansoni has been verified only in vitro, but lack of the validation in vivo. In animals, due to complex environmental factors, such as drug absorption, etc., does it still have obvious killing effect on S. mansoni? In addition, does the drug damage animals? What’s the side effect of LSD1 inhibitor on animals? I think the manuscript should be included this part. If the LSD1 inhibitor causes severe damage to the host, this conclusion needs to be adjusted here.

2.Line 285-286: This part about adult worm motility and egg laying assay should use the paired adult worms, but not mixing with paired or unpaired.

3.What is the statistical method used in the article? Please add in the part of method.

4.How many parasites have been observed in each group in figure 5, 6 and 7? Are all the same in each group? Can you provide the statistical results?

PLOS authors have the option to publish the peer review history of their article (what does this mean?). If published, this will include your full peer review and any attached files.

Reviewer #1: No

Reviewer #2: No

Reviewer #3: No

---

## [Decision Letter · Decision Letter 1]

28 Apr 2020

Dear Dr. Prof. Fantappié,

We are pleased to inform you that your manuscript 'Pharmacological inhibition of lysine-specific demethylase 1 (LSD1) induces global transcriptional deregulation and ultrastructural alterations that impair viability in Schistosoma mansoni' has been provisionally accepted for publication in PLOS Neglected Tropical Diseases.

Best regards,

Timothy G. Geary, PhD

Deputy Editor

Timothy Geary

Deputy Editor

Reviewer's Responses to Questions

**Key Review Criteria Required for Acceptance?**

**Methods**

-Are the objectives of the study clearly articulated with a clear testable hypothesis stated?

-Is the study design appropriate to address the stated objectives?

-Is the population clearly described and appropriate for the hypothesis being tested?

-Is the sample size sufficient to ensure adequate power to address the hypothesis being tested?

-Were correct statistical analysis used to support conclusions?

-Are there concerns about ethical or regulatory requirements being met?

Reviewer #1: Authors have adequately addressed the concerns raised by the reviewers

Reviewer #2: (No Response)

Reviewer #3: (No Response)

**Results**

-Does the analysis presented match the analysis plan?

-Are the results clearly and completely presented?

-Are the figures (Tables, Images) of sufficient quality for clarity?

Reviewer #1: Authors have adequately addressed the concerns raised by the reviewers

Reviewer #2: (No Response)

Reviewer #3: (No Response)

**Conclusions**

-Are the conclusions supported by the data presented?

-Are the limitations of analysis clearly described?

-Do the authors discuss how these data can be helpful to advance our understanding of the topic under study?

-Is public health relevance addressed?

Reviewer #1: Authors have adequately addressed the concerns raised by the reviewers

Reviewer #2: (No Response)

Reviewer #3: (No Response)

**Editorial and Data Presentation Modifications?**

Reviewer #1: (No Response)

Reviewer #2: (No Response)

Reviewer #3: (No Response)

**Summary and General Comments**

Reviewer #1: Authors have adequately addressed the concerns raised by the reviewers

Reviewer #2: (No Response)

Reviewer #3: (No Response)

PLOS authors have the option to publish the peer review history of their article (what does this mean?). If published, this will include your full peer review and any attached files.

Reviewer #1: No

Reviewer #2: No

Reviewer #3: No

---

## [Editor Report · Acceptance letter]

2 Jun 2020

Dear Dr. Prof. Fantappié,

We are delighted to inform you that your manuscript, "Pharmacological inhibition of lysine-specific demethylase 1 (LSD1) induces global transcriptional deregulation and ultrastructural alterations that impair viability in Schistosoma mansoni," has been formally accepted for publication in PLOS Neglected Tropical Diseases.

Best regards,

Serap Aksoy

Editor-in-Chief

Shaden Kamhawi

Editor-in-Chief
